ecology, health and disease and epidemiology

*Desmodus rotundus*, Chiroptera, metapopulation maintenance, zoonoses, eco-immunology, wildlife

**Author for correspondence:**
Diana K. Meza
e-mail: d.villa-meza.1@research.gla.ac.uk

†Joint senior authors.

# Ecological determinants of rabies virus dynamics in vampire bats and spillover to livestock

Diana K. Meza[1,2], Nardus Mollentze[1,2], Alice Broos[1,2], Carlos Tello[3,4], William Valderrama[3,5], Sergio Recuenco[6], Jorge E. Carrera[7,8], Carlos Shiva[5], Nestor Falcon[5], Mafalda Viana[1,†] and Daniel G. Streicker[1,2,†]

[1]School of Biodiversity, One Health and Veterinary Medicine, University of Glasgow, Glasgow, UK
[2]Medical Research Council-University of Glasgow Centre for Virus Research, Glasgow, UK
[3]ILLARIY (Asociación para el Desarrollo y Conservación de los Recursos Naturales), Lima, Perú
[4]Yunkawasi, Lima, Perú
[5]Universidad Peruana Cayetano Heredia, Lima, Perú
[6]Facultad de Medicina San Fernando and [7]Departamento de Mastozoología, Museo de Historia Natural, Universidad Nacional Mayor de San Marcos, Lima, Perú
[8]Programa de Conservación de Murciélagos de Perú, Perú

DKM, 0000-0001-9796-6706; NM, 0000-0002-2452-6416; AB, 0000-0001-7593-1000; CT, 0000-0003-3758-265X; WV, 0000-0003-4626-1444; SR, 0000-0002-8446-7411; JEC, 0000-0001-6644-4518; CS, 0000-0002-8473-0128; NF, 0000-0003-4144-0494; MV, 0000-0001-5975-6505; DGS, 0000-0001-7475-2705

The pathogen transmission dynamics in bat reservoirs underpin efforts to reduce risks to human health and enhance bat conservation, but are notoriously challenging to resolve. For vampire bat rabies, the geographical scale of enzootic cycles, whether environmental factors modulate baseline risk, and how within-host processes affect population-level dynamics remain unresolved. We studied patterns of rabies exposure using an 11-year, spatially replicated sero-survey of 3709 Peruvian vampire bats and co-occurring outbreaks in livestock. Seroprevalence was correlated among nearby sites but fluctuated asynchronously at larger distances. A generalized additive mixed model confirmed spatially compartmentalized transmission cycles, but no effects of bat demography or environmental context on seroprevalence. Among 427 recaptured bats, we observed long-term survival following rabies exposure and antibody waning, supporting hypotheses that immunological mechanisms influence viral maintenance. Finally, seroprevalence in bats was only weakly correlated with outbreaks in livestock, reinforcing the challenge of spillover prediction even with extensive data. Together our results suggest that rabies maintenance requires transmission among multiple, nearby bat colonies which may be facilitated by waning of protective immunity. However, the likelihood of incursions and dynamics of transmission within bat colonies appear largely independent of bat ecology. The implications of these results for spillover anticipation and controlling transmission at the source are discussed.

## 1. Introduction

Understanding the determinants of pathogen transmission within bat populations is necessary to manage health risks to human and companion animals as well as threats to bat conservation [1]. Unfortunately, data from bat populations typically arise from short-term studies that are unlikely to disentangle the ecological complexity behind pathogen circulation within bats or spillover to other species [2]. Moreover, the health surveillance data that have been instrumental to understand human and livestock diseases are limited to bat

pathogens, including zoonoses, since detected spillover events are typically infrequent [1,3]. The currently limited understanding of pathogen transmission in bats frustrates strategic interventions for disease control, including vaccination of spillover hosts or efforts to reduce incidence within bat populations by population control (i.e. culling), bat dispersal or novel therapeutics [4–8]. Longitudinal monitoring of pathogen incidence has been suggested as a possible approach to understand pathogen circulation in bats. However, examples remain scarce due to the logistical constraints implicit in longitudinal studies of reclusive wildlife [1] and when achieved, have left considerable uncertainty in the mechanisms underlying pathogen maintenance, perhaps due to the limited breadth of data available [9,10]. Past studies have further been limited by low recapture rates, which preclude understanding the long-term fate of infected individuals. Biological systems with temporally and/or spatially extensive data, and where the infection status individual bats can be tracked through time, would be valuable to assess the scope for pathogen monitoring to inform management.

In Latin America, vampire bat (*Desmodus rotundus*) transmitted rabies virus (VBRV) causes sporadic lethal infections in humans and annual losses in the tens of millions of dollars from livestock mortality [11]. Despite over 50 years of investment in vaccination of humans and livestock, and culling of bats using anti-coagulant poisons, the burden and geographical range of VBRV is increasing in several countries [12]. Further, observational studies and epidemiological models suggest that bat culls might inadvertently enhance rabies transmission by increasing mixing of bat colonies, analogous to dynamics observed for bovine tuberculosis in culled badgers in the United Kingdom [4,13]. Although the long-term maintenance strategy of VBRV is among the best understood for bat-associated zoonoses, uncertainties and untested assumptions remain (electronic supplementary material, table S2). For example, epidemiological [14–16] and phylogenetic analyses of outbreaks in livestock showed travelling waves of infection at landscape scales, which led to patterns of lineage invasion, extinction and replacement at local levels [14,15,17]. Consistent with spatially mediated maintenance, mathematical models based on serological data from bats predicted that single colonies cannot sustain transmission over long time periods [10]. However, the frequency, duration and magnitude of local epizootics, and whether variation in these epidemiological outcomes varies predictably with environmental factors or bat demography remain unknown [15,16]. Mathematical models also predicted that observed patterns of exposure would require waning protective immunity from sub-lethal exposures [10]. Although virus-neutralizing antibodies are commonly observed, neither their protective nature nor whether they wane predictably in wild bats has been empirically verified [4,18]. Despite circumstantial evidence for spatially mediated maintenance, an alternative hypothesis is that VBRV persists within bat colonies via mechanisms that are currently poorly understood. For example, stress has been hypothesized to induce the reactivation and shedding of other bat RNA viruses [19,20]. While stress-induced activation of VBRV has not been experimentally evaluated, unexpected outbreaks in bats brought into captivity were speculated to be linked to stress [21,22]. Alternatively, prolonged incubation periods are well recognized in rabies and might enable epizootics following recovery of susceptible populations to threshold levels, generating patterns that might outwardly resemble extinction–recolonization dynamics driven by viral spatial spread [16,23]. Clarifying knowledge gaps surrounding VBRV maintenance might improve allocation of vaccines to spillover populations at greatest risk and guide strategies for vampire bat management [24,25]. For example, if VBRV is maintained locally, understanding environmental stressors on bats and managing susceptible bat populations below epizootic thresholds (by vaccination or culling) might mitigate its burden [12,16]. Alternatively, if incidence in bats and spillover arise through processes occurring at landscape scales, which are less dependent on local bat density [4,10], then understanding the geographical scale of maintenance would enable synchronization of management across enzootic areas.

Resolving VBRV circulation within bat populations has been challenging because of the limited data available. Active infection is rare in wild bats (less than 1% infected), precluding direct insights into population-level incidence through RT-PCR and viral phylogenetic studies [26]. Incidence and molecular data from infected livestock can be extensive, but observation biases inherent to passive surveillance (e.g. variable reporting and vaccination coverage) mean these data provide only indirect insights into the dynamical processes within bat populations. Serology, the detection of pathogen-specific antibodies in blood serum, therefore remains the most practical option to monitor VBRV in wild bats. Indeed, experimental inoculations suggest that virus neutralizing antibodies are detectable for less than six months, providing a coarsely time-resolved metric of viral exposure which should approximate transmission at the population scale [27]. Here, we used data from longitudinal (11 years) serological monitoring of 39 vampire bats colonies in Peru to study the ecological mechanisms that enable long-term rabies maintenance. Specifically, we aimed to: (1) identify the nature and geographical scale of viral maintenance; (2) evaluate whether individual, population, landscape or spatio-temporal variables explain seroprevalence; and (3) empirically test whether viral exposures provide transient protective immunity. Finally, (4) using a dataset of co-occurring rabies outbreaks in sympatric livestock, we assess whether antibody prevalence in bats is a useful proxy for spillover risk to non-bat species.

## 2. Methods

### (a) Field capture and study design

Between 2007 and 2017, we sampled 39 vampire bat colonies from Peru (figure 1a; electronic supplementary material, text). For each colony, the presence/absence of non-vampire bat species was recorded (sites with only *D. rotundus* = 48%). For each individual, we recorded sex, age group and reproductive status, and collected blood samples to obtain serum [4]. Descriptions of bat handling are provided in the electronic supplementary material.

For visualization and statistical purposes (to have robust sample size in each site), data from 33 bats from 6 poorly sampled colonies (defined as those with less than 10 individuals) were joined with other colonies within 5 km [28], creating a total of 33 bat colonies (hereafter, 'bat sites'). The minimum distance between joined colonies was 0.23 km. We considered this aggregation epidemiologically appropriate since vampire bats often use multiple roosts within geographically nearby areas [28] and previous studies have suggested epidemiological linkage of colonies separated by up to 8 km [18]. Although it is possible that colonies at this scale are not directly connected due to heterogeneities in bat dispersal linked to colony type (e.g. bachelor versus maternity

*Proc. R. Soc. B* **289**: 20220860

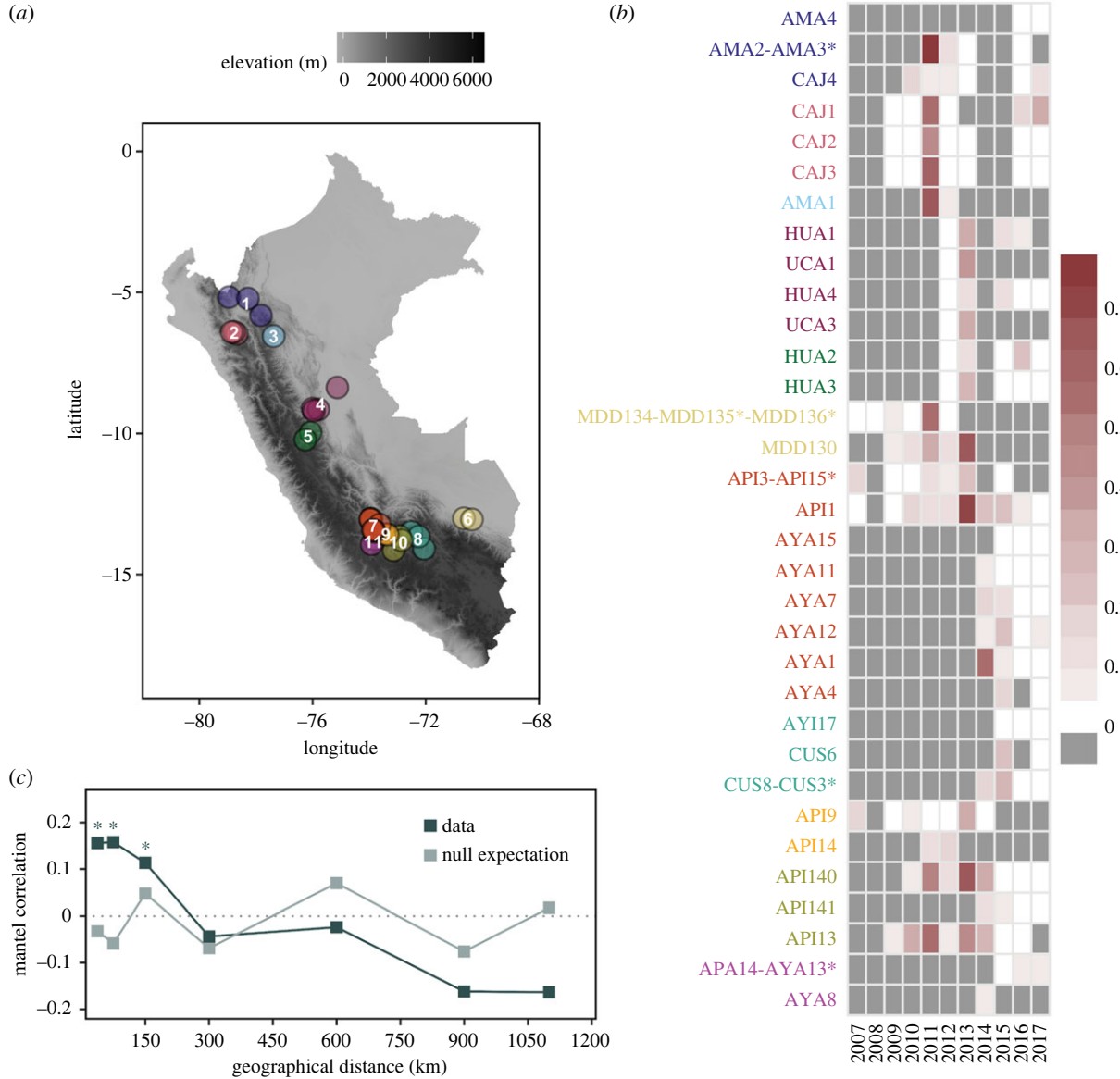

**Figure 1.** Observed rabies seroprevalence in vampire bats in Peru. (*a*) Map of Peru showing the location of the bat sites sampled. Colours represent spatial clusters, created using hierarchical clustering of least-cost-distances based on elevation. The grey ramp indicates the elevation gradient in metres. (*b*) Heatmap of rabies seroprevalence (red colour ramp) in vampire bats across years (columns) and sites (rows; colour coded by cluster as in figure 1*a*). Asterisks in the names show colonies that had ≤10 bats sampled and were joined to a nearby colony (≤5 km). No data are represented in grey. (*c*) Correlogram of the spatial synchrony across distance classes. The lighter colour represents the null expectation from randomized data. Asterisks denote significant correlations. (Online version in colour.)

colonies), we verified that sex and age ratios were comparable in colonies that were grouped (electronic supplementary material, table S1). To analyse correlates of rabies seroprevalence (see below), we hierarchically clustered the 33 bat sites into 11 clusters using pairwise distances (see landscape metrics methods below; electronic supplementary material, figure S4–S6; figure 1*a*). The minimum Euclidean distance between clusters was 56.61 km (clusters 9 and 10). Since these sites are part of a long-term project, two subsets of the data were previously published [4,26]. Here, we expand the number of sites and years of data to investigate dynamics that previous analyses were unable to address.

### (i) Detection of rabies virus neutralizing antibodies
Sera from 2007–2010 were tested for rabies virus neutralizing antibodies (RVNAs) using a micro-neutralization rapid fluorescent focus inhibition test (micro-RFFIT) [4]. Sera from 2011–2017 were tested using a pseudotype micro-RFFIT (pmRFFIT) which avoided use of live rabies virus and enabled direct quantification of RVNA titres [29]. We used a cutoff of 0.166 IU ml$^{-1}$ to define

seropositivity on tested samples by pmRFFIT which was previously shown to maximize the balance of sensitivity and specificity [29]. Serological results in tested samples by both methods ($n = 2365$) are 93.7% similar [29].

### (ii) Landscape metrics of habitat suitability, spatial isolation and rabies presence
We designed four landscape variables to estimate the suitability and spatial isolation of each vampire bat site. Unless otherwise noted, each variable was averaged within a 10 km radius around each site using the 'raster' package of R v. 4.0.3 [30,31]. Explanations of how each factor might be linked to seroprevalence are explained in electronic supplementary material, table S2. Landscape variables included: (1) 'suitability', calculated as the proportion of neighbouring cells assumed to be suitable for vampire bat presence (less than or equal 3600 m [17]) using CGIAR-SRTM 90-m resolution elevation data [32]; (2) 'terrain ruggedness index' (TRI), which is the relative difference between the elevation of a central cell and its eight surrounding cells [31]

(slope and roughness were also computed but were highly correlated with TRI; electronic supplementary material, figure S1); (3) 'livestock density,' estimated as the average densities of horses, goats, pigs, sheep and cattle within 10 km of each bat site using the 2010 Gridded Livestock of the World, the highest resolution data available on livestock densities (5 min of arc resolution [33]); and (4) a 'least-cost distance' (LCD) measure of spatial isolation, assuming a model where movement costs increase linearly with elevation until 3600 m, then movement cost is infinite (i.e. a hard barrier to bat dispersal) [24]. We averaged LCDs from 20 randomly selected points located 40 km away (Euclidian distances) from each site. Considering that VBRV has been reported to spread at velocities of approximately 10 km year$^{-1}$ (range = 9.1 to 17.2 km year$^{-1}$) in Peru [17], we developed additional variables to describe how the reported incidence of VBRV in livestock at two distances from bat sites would affect future seroprevalence in bats [34]. The 'inner-circle' was calculated as the number of outbreaks occurring up to 10 km from the site in the 12-month period prior to bat sampling. The 'outer-circle' was the number of outbreaks occurring between 10 and 20 km from the site ≥12 months and ≤36 months prior sampling (24-month window). The inner and outer circle variables were generated from a national database of confirmed rabies-positive animals, provided by the National Service of Agrarian Health of Peru.

## (b) Statistical analysis

### (i) Spatial synchrony in seroprevalence
We used pairwise measures of spatial synchrony in seroprevalence to understand the biological–geographical scale over which patterns of rabies transmission were correlated (expecting non-independence in bat colonies at shorter distances). Specifically, for all pairs of sites with greater than 2 years of data (29 out of 33), we used Mantel tests (from 'vegan' package [35]) to analyse the relationship between the geographical distance between sites (Euclidian distances) and their observed synchrony in seroprevalence, calculated as Spearman correlations in annual seroprevalence. To visualize correlations at shorter distances, we plotted a correlogram with different distance classes. We compared observed relationships between geographical distance and seroprevalence synchrony to a null expectation, generated by randomizing observations of seroprevalence at the site and year level.

### (ii) Ecological correlates of rabies seroprevalence
To evaluate whether individual, population, landscape or spatiotemporal variables explain seroprevalence, binomial generalized additive mixed models (GAMMs) were fit via restricted maximum likelihood (REML) to individual bat seropositivity (0/1) using the 'mgcv' package [36]. We chose GAMMs as they offer a robust framework to model complex potential nonlinear effects [37]. However, since nonlinear effects in GAMMs can lead to overfitting and reduce computational efficiency, we carried out an initial analysis to identify which continuous variables required smoother effects. Specifically, we fit a GAMM that included all potential explanatory categorical and continuous variables, where penalized smoothers were used for all continuous variables. Variables with effective degrees of freedom (EDFs) > 1 were retained as smoothers in the subsequent analyses and others modelled as linear. Most continuous covariates were tested as penalized thin plate regression splines, starting with the default basis dimensions from 'mgcv' as recommended by [36,37]. For month, we used a cyclic cubic smoother, specifying months 2 and 12 as the start and endpoints (there were no data for month 1) and we included an interaction between year and cluster, where we used a factor-smooth interaction. Consequently, our initial model included: sex, age, reproductive status, presence/absence of other bat species, as linear effects; TRI, LCDs, elevation, livestock density, inner-circle, outer-circle outbreaks,

month and year-cluster interaction as smoother effects. Among the continuous variables considered, elevation, inner-circle, month and the interaction year-cluster were retained as smoothers (electronic supplementary material, table S3).

We next compared potential candidate random effects structures using the Akaike information criterion (AIC) and deviance explained (Dev.Exp; electronic supplementary material, table S4). The most competitive random effect structure (site alone) was identified as the minimal adequate model that described rabies seroprevalence. The comparison of all possible variable combinations (including models with a single fixed effect) was done through *dredge* from the 'MuMIn' package [38]. We calculated the relative deviance as a measure of goodness of fit as in [39]. After model selection, we adjusted the number of basis functions of the smoothers in the top model using *gam.check* from 'mgcv' to confirm these were still appropriate [36].

We hypothesized that the presence of males and juveniles, roost suitability, livestock density, other bat species, inner-circle and outer-circle would increase rabies seroprevalence by altering bat density and high-risk (including interspecific) contacts, that LCD would decrease rabies seroprevalence by epidemiologically isolating colonies and that TRI might either increase seroprevalence if it correlates with nearby roost availability or decrease seroprevalence if it isolates colonies. We expected to observe seasonality in seroprevalence given reports of birth seasonality [23,40] and that colonies within the same cluster would have similar rabies exposure dynamics through time due to epidemiological connectivity arising through bat dispersal.

### (iii) Dynamics of antibody loss and seroconversion
We used data from 427 recaptured individuals (a total of 486 recapture events with bats recaptured 1 to 5 times) to explore patterns of antibody waning and seroconversion (electronic supplementary material, figure S8). Recaptures with changes in serological status were classified as 'negative-to-positive', indicative of rabies exposure in the time between captures; or 'positive-to-negative', indicative of waning of a previously detectable antibody titre. Recaptures with no serological category change were classified as: 'negative-to-negative' or 'positive-to-positive'. We considered each recapture event of an individual as independent (i.e. capture 1 to 2 and capture 2 to 3, etc.). We compared the distributions of times to recapture across the four event classes to assess whether sublethal exposures provide transient protective immunity. Specifically, if positive-to-negative recapture intervals are longer than negative-to-negative intervals, it would suggest enhanced survival due to protective immunity against rabies. Shorter positive-to-positive intervals relative to other classes would be consistent with antibody waning and the transient nature of protective immunity. We tested differences among event classes using a Kruskal–Wallis test, using recapture interval as the response variable and the class of serological transition as a covariate. Pairwise differences between classes were tested using a *post-hoc* Dunn test. To investigate patterns of titre loss, we plotted the titres and seropositivity status of recaptures for which continuous VNA titres were available (n = 211; 2011–2017) over time.

### (iv) Association between seroprevalence in bats and spillover to livestock
We used generalized linear mixed models (GLMM) with Poisson distributions to test whether bat seroprevalence was correlated with rabies outbreaks in livestock (electronic supplementary material, table S2). Separate models described the number of rabies outbreaks in livestock within 10 km of each bat site at three temporal scales: 6, 12 and 24 months. Since RVNAs are detectable for days to months after infection, RVNAs observed in bats can reflect heightened rabies circulation in the past or present [27], leading to spillovers to livestock before or after

the bat sampling date. Consequently, temporal scales were centered on the bat sampling point (e.g. the 6-month window contained outbreak data from three months before and after the bat sampling point). Local livestock density was included as a predictor to test hypothesized effects of livestock on rabies outbreaks. Specifically, as the primary food source for vampire bats, livestock are reported to increase bat density and therefore possibly rabies incidence [41]. Livestock density may also influence the probability that outbreaks are detected or reported [42]. Random effects included bat site and an individual identifier per seroprevalence observed (i.e. the site and date surveyed) which reduced overdispersion. To assess the goodness of fit of these models, we calculated the conditional and marginal pseudo $R^2$ using the delta method [43]. We further examined the correlation of seroprevalence in bats with rabies outbreaks over larger spatio-temporal scales by repeating GLMMs using data re-calculated at 20 and 30 km from each site.

## 3. Results

Over 11 years (2007–2017), 4889 bats were captured in the 33 sites. Serological information was obtained for 4196 serum samples, collected from 3709 individuals. Seroprevalence across all sites and years sampled was 15.22% (95% CI: 14.13, 16.31), but varied considerably across years, from 4.03% (95% CI: 2.44, 5.61) to 44.78% (95% CI: 40.40, 49.15) (electronic supplementary material, figure S2). Site seroprevalence across years varied from 0% (observed in 30 out of 138 sampling instances with greater than 10 sampled bats) to 73.68% (electronic supplementary material, figure S3). We also observed peaks of seroprevalence alternating with periods of 0% or very low seroprevalence, suggesting epizootic dynamics with potential viral extinctions. In other sites seroprevalence fluctuated at levels greater than 0% throughout the sampling period, consistent with continuous, but time-varying viral exposures (figure 1b).

### (a) Spatial synchrony in seroprevalence

Synchrony in bat seroprevalence declined with increasing geographical distance between sites (Mantel $r = 0.32$, $p = 0.001$; null-expectation: Mantel $r < 0.001$, $p = 0.4$) and was significant only at the shortest distance classes evaluated (Mantel $r = 0.16$, $p \leq 0.02$; figure 1c). Sites separated by greater than 150 km were not significantly correlated ($p > 0.05$ figure 1c). This suggests VBRV circulates as distinct asynchronous cycles that are compartmentalized to different geographical regions.

### (b) Ecological variables associated with rabies seroprevalence

Our model selection identified 13 competitive models ($\Delta$AIC < 2, out of greater than 10 000 models). Each competitive model explained approximately 28% of deviance, predominately through four variables: year-cluster interaction (Dev.Exp = 16.94% in the top model), inner-circle outbreaks (Dev.Exp = 11.80%), month (Dev.Exp = 11.49%) and the random effect of site (Dev.Exp = 11.20%) (electronic supplementary material, figure S7). The top 13 models also contained a weak effect of livestock density, explaining less than 0.5% of deviance. The remaining retained variables, which defined the differences among the competitive models, explained only trivial deviance. Thus, while we identified multiple statistically competitive models, they supported equivalent biological conclusions. We

therefore focus on the results from the top model (Dev.Exp = 27.96%, d.f. = 75.35)

Consistent with our synchrony analysis, the year–cluster interaction revealed different temporal dynamics across clusters and correlated patterns within clusters (figure 2a), suggesting independent VBRV enzootic cycles across regions. The 'inner-circle' effect showed that increasing numbers of reported rabies outbreaks in nearby livestock was associated with higher seroprevalence in bats up to about 9 outbreaks (log-odds = 1.49 [95% CI: 0.7,2.2]) compared to when no nearby outbreaks were reported. Beyond 9 outbreaks, predicted seroprevalence in bats fluctuated inconsistently, reflecting the sparsity of bat sampling observations with very large numbers of rabies outbreaks within the defined spatial and temporal window of surveillance (figure 2b). The relationship between month and seroprevalence showed a trough at the start of the dry season (April) and 2 possible peaks during other times of the year (figure 2c). Curiously, livestock density had a negative effect on seroprevalence (log odds=−0.46 [95% CI: −0.79, −0.14]), inconsistent with the expectation of increased rabies transmission due to the positive effect of livestock on bat density. Importantly, no environmental or bat demographic variable was consistently retained by our model selection (electronic supplementary material, figure S7). Together, these results suggest that patterns of rabies exposure arise through spatio-temporal processes operating at landscape scales, but that the likelihood or dynamics of epizootics within bat colonies are not influenced by any measured aspect of bat ecology, including proxies for bat density and age structure that would be expected to reflect the availability of susceptibles.

### (c) Dynamics of antibody loss and seroconversion

From 486 recaptures with serological data, 14.6% showed seroconversion from negative-to-positive and 11.9% from positive-to-negative (figure 3a). Relatively few bats (3.91%) were positive-to-positive across recaptures. Recapture intervals differed across seroconversion classes (figure 3a; $\chi^2 = 23.96$, d.f. = 33, $p < 0.001$) and were significantly longer in bats that seroconverted from positive-to-negative than in any other class (1.33 years versus: median negative-to-negative = 1; median negative-to-positive = 0.91). The shortest recapture intervals occurred for bats that remained seropositive (median = 0.67 years), consistent with antibody waning (i.e. bats recaptured at longer intervals were instead positive-to-negative). RVNA titres declined in 67.3% of recaptured individuals with continuous titre information, indicating long-term survival of rabies-exposed bats and antibody waning (figure 3b). Together, the extended recapture intervals from the positive-to-negative serological transition and the waning of RVNA titres support the hypothesis that sublethal exposures provide transient protective immunity.

### (d) Association between seroprevalence in bats and spillover to livestock

The number of rabies outbreaks in livestock was correlated with seroprevalence in bats across all time windows ($p < 0.05$) but not nearby livestock density ($p > 0.1$, figure 4). Even though seroprevalence explained part of the variation of the spillovers, the random effect of the site explained most of the variability, as evidenced by the high conditional $R^2$ of

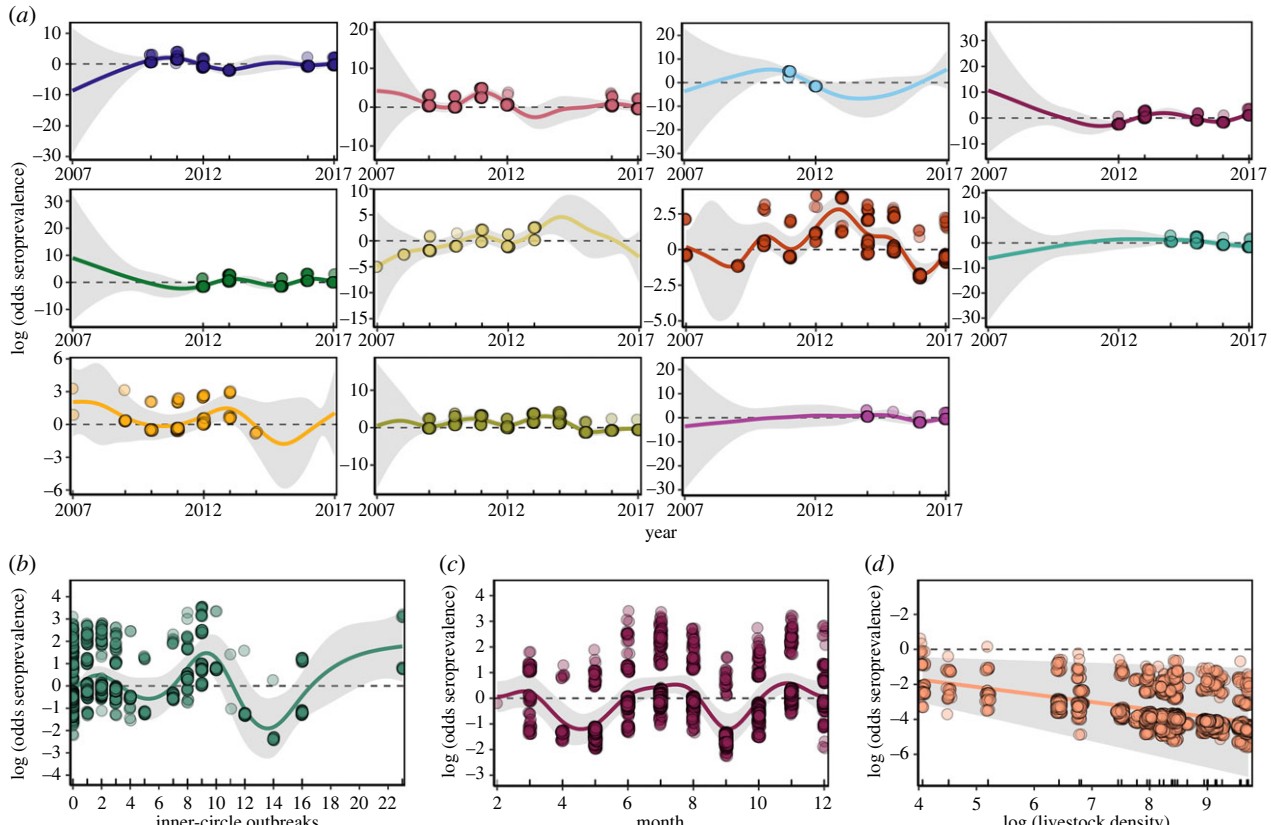

**Figure 2.** Predicted effects from the top model on rabies seroprevalence in vampire bats. Ticks along the *x*-axis indicate field-derived observations. Points correspond to the partial residuals and lines to the predicted effect of each variable. Grey shaded areas represent 95% CIs. (*a*) Effects of the interaction between year and cluster. Each panel represents a cluster, colours as in figure 1*a*. (*b–d*) Predicted effects of the total number of outbreaks occurring up to 10 km from a site in a 12-months period prior to bat sampling, sampling month and livestock density. (Online version in colour.)

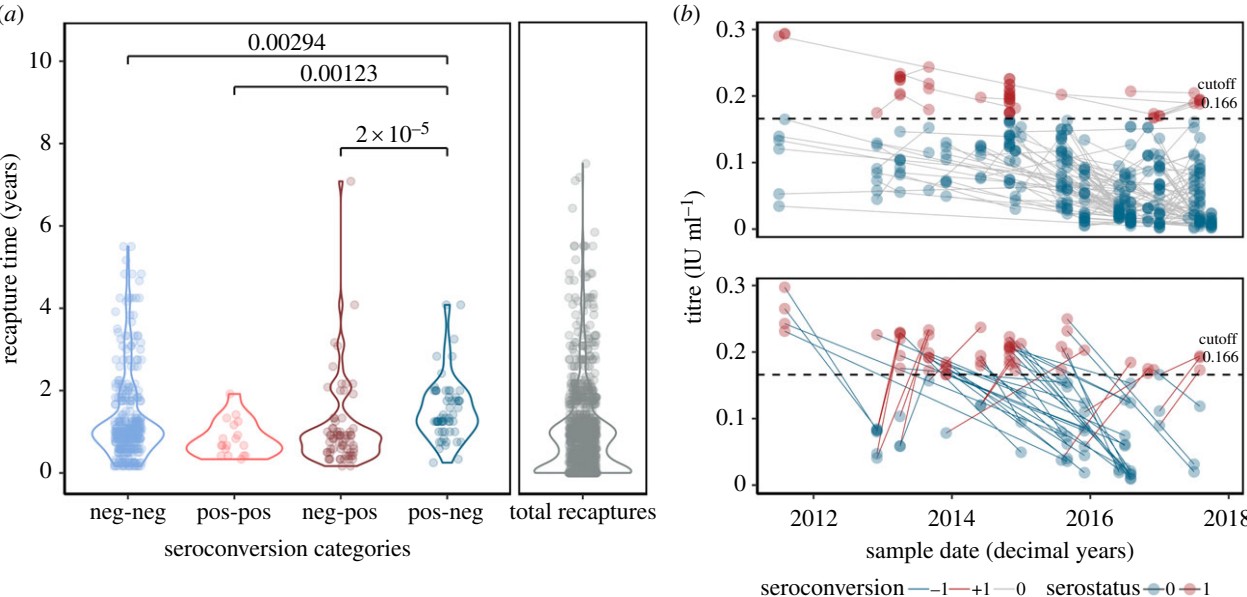

**Figure 3.** Seroconversion of recaptured bats. (*a*) Density of the observed times between recaptures for each seroconversion class, and total recaptures (grey). Significant pairwise differences are shown ($p < 0.05$). (*b*) Change in rabies neutralizing antibody titres over time. Lines connect individuals with the direction of seroconversion in different colour (top panel no change and bottom panel change). Points represent the date of (re)capture with the serostatus at that time in different colours. The dashed line represents the cutoff defining seropositivity based on [29]. (Online version in colour.)

full models ($R^2 > 0.86$) compared to the low marginal $R^2$ for the bat seroprevalence effect ($R^2 < 0.07$). Models using 20 and 30 km radii were overdispersed (electronic supplementary material, figure S9), but similarly indicated positive relationships between bat seroprevalence and rabies spillover to livestock (electronic supplementary material, figure S10).

## 4. Discussion

Longitudinal monitoring has been proposed as a route to clarify the individual and population level determinants of infection in bats but has only rarely been accomplished [1,2]. By studying 39 Peruvian vampire bat colonies over 11

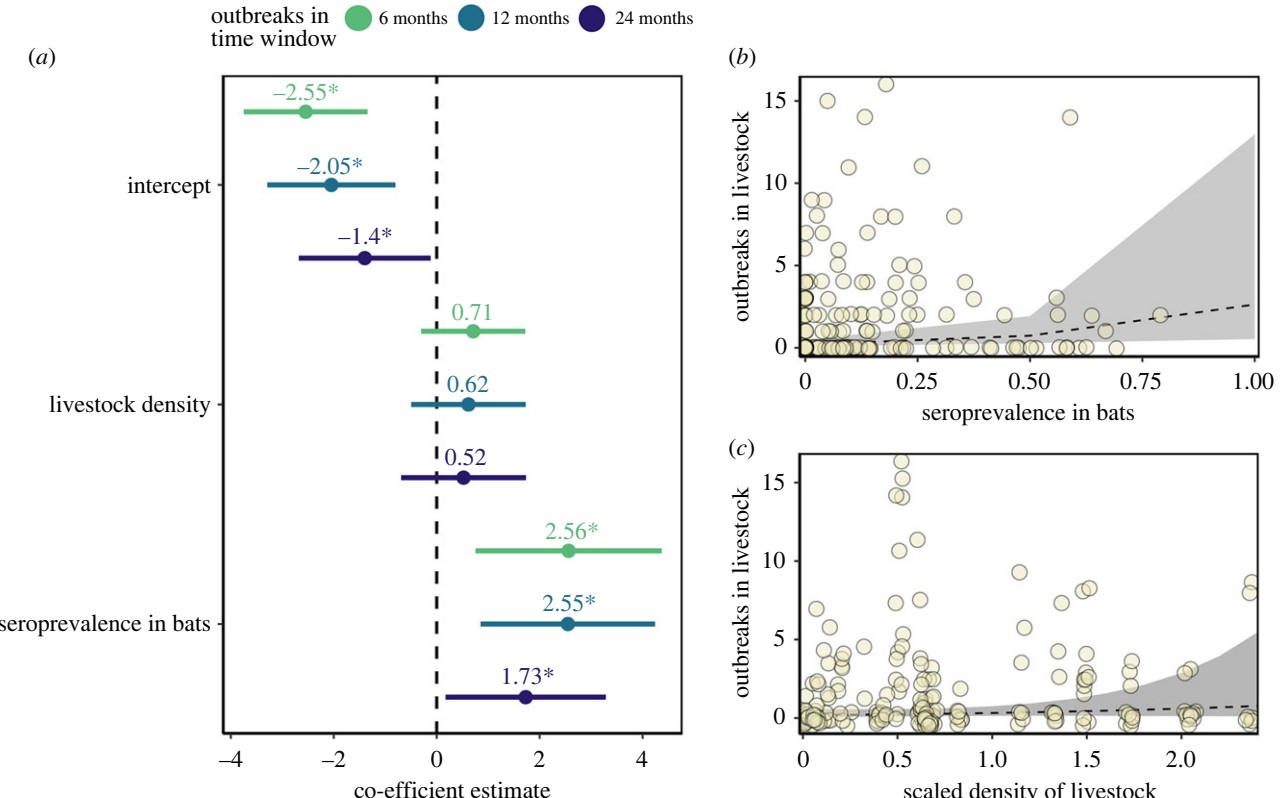

**Figure 4.** Effects of bat seroprevalence on the intensity of rabies spillover to livestock. (*a*) Mean coefficient estimates with associated 95% CIs (*x*-axis) of the covariates (*y*-axis) used for each of the three models fitted to different temporal scales (colours; asterisk *p* < 0.05). (*b,c*) Observed and predicted number of outbreaks (*y*-axis) associated with seroprevalence in bats and livestock density (*x*-axis) in the 10 km – 12-month model. Circles represent observations and shaded areas represent 95% CIs for estimations. (Online version in colour.)

years, we showed that the spatial and temporal distribution of rabies exposure is dynamic in space and time and synchronized across short distances, but largely independent of landscape characteristics, bat demography or livestock density. Monitoring recaptured bats showed long-term survival of rabies-exposed bats and supported the hypothesis that sublethal infections provide transient, protective immunity. The complex interplay between viral spatial spread driven by bat dispersal and a dynamic immunological landscape appear to together determine rabies incidence in bat colonies. Finally, seroprevalence in bats was positively correlated with rabies incidence in livestock, but considerable unexplained variation highlighted the exceptional challenge of anticipating spillover of a spatio-temporally dynamic virus.

Decaying synchrony in seroprevalence with distance, asynchronous patterns in viral exposure observed among clusters of bat sites, putative viral extinctions and re-invasions in individual sites, and effects of nearby viral circulation on future exposures (figures 1*c* and 2*a*) are broadly consistent with viral invasions across the landscape, which are followed by variable durations of localized transmission. This phenomenon has been suggested from reports of VBRV in livestock, but never previously observed within wild bat populations [14,15,17]. Although our larger scale measure of this invasion process (outer-circle describing patterns of livestock outbreaks at 10–20 km from each site) did not predict seroprevalence in bats, this may reflect an inevitable effect of observing larger geographical areas. For example, areas far from observed bat sites may have neither livestock nor vampire bat populations, whereas these necessarily co-occur at shorter distances, explaining why the inner-circle variable, but not the outer

circle variable was correlated with seroprevalence. Interestingly, seroprevalence also exceeded zero over consecutive years in some sites, suggesting sustained viral exposures (figure 1*b*). Given that livestock rabies outbreaks tend to have prolonged inter-epizootic periods [15], we suggest that sustained seropositivity within bat colonies is most parsimoniously explained by occasional viral exposures driven by dispersal of infected bats between sites, rather than by permanent viral circulation within certain colonies [34]. Given the absence of demographic or landscape predictors of seroprevalence, apparent sustained seropositivity versus apparent extinctions likely depends more on the spatial configuration of vampire bat colonies than their size, sex ratios or age composition. Therefore, improving the currently sparse knowledge of the spatial distributions of vampire bat colonies should be a priority which could eventually allow anticipation of whether rabies will be maintained or go extinct within an area, improving allocation of resources for prevention and control.

Our results also provided some evidence for seasonality in vampire bat rabies, with seroprevalence declining at the beginning of the dry season (April–May; figure 2*c*). Such effects might be associated with bat behaviour and/or birth seasonality. Dry seasons in other countries in South America are associated with changes in foraging behaviour and in the time spent in roosts, which might alter viral exposure rates [16,23]. Higher seroprevalence was observed during most of the rainy season. This could reflect either dispersal of rabies-naive juveniles (since greater mixing is expected to increase seroprevalence), or yet to be described seasonal changes in bat behaviour [10,23]. Seasonal spillover to livestock has not been widely detected, further suggesting that

individual factors (e.g. seasonal reproduction in vampire bats) might not be enough to cause or enhance spillovers [14,17,23]. By contrast, seasonality might be more relevant to between-colony transmission. In this scenario, the observed seroprevalence is the result of exposure of resident bats to dispersing infected bats, triggering abortive infections and the production of detectable antibodies. Since abortive infections are common and spillovers are probably rare [27], such exposures might occur without triggering local epizootics large enough to lead to spillovers.

Given that livestock are the primary food source for vampire bats [42], vampire bat populations tend to be larger in livestock rearing areas [4,41], leading to the expectation of higher levels of viral circulation in areas with greater livestock density. Intriguingly, we found no evidence for this anthropogenic amplification of rabies dynamics. Areas with higher livestock density did not have larger numbers of rabies outbreaks (figure 4). Although it is conceivable that the absence of this relationship reflects higher vaccination rates where livestock are more numerous, this is unlikely given the reactive vaccination practices (i.e. after outbreaks are detected) that are common in Peru [11]. We also found that seroprevalence in bats appeared to decline (albeit weakly) rather than increase with livestock density (figure 2d). If our assumption that livestock density increases bat population density, as observed elsewhere, holds in our study sites, these findings imply that rabies transmission only weakly depends on bat density, supporting earlier concerns over the utility of culling bats for rabies control [4,10]. Weak density dependence may arise if biting (the main transmission route of VBRV among bats) is independent of bat density and/or the ephemeral nature of viral circulation within individual bat colonies arising from metapopulation maintenance. A possible caveat to this result is that bat colonies in areas with greater livestock density might also experience more culling and therefore have lower density. However, the reactive nature of bat culls to rabies outbreak in Peru and earlier work suggesting a positive, rather than a negative influence of culling on rabies exposures in bats which would be inconsistent with this scenario [4]. Why seroprevalence not only failed to increase but also appeared to decline with livestock density is unclear. One possibility is that increases in food resources might require less foraging effort and lower nutritional stress, enabling higher immune investment [19,41].

Our results suggest ecological explanations of why current rabies control methods have had limited success and offer opportunities for management of vampire bat rabies. In theory, culls would be more effective for enzootic pathogens with density-dependent transmission [44]. The results shown here suggest that neither of these prerequisites may hold. Indeed, given spatially driven viral maintenance, successful use of culling for rabies control would require culls that are strategically deployed using landscape-level epidemiological knowledge (i.e. culling in advance of viral invasion) or spatially synchronized across enzootic areas. While the former strategy is plausible in rare situations where the pathways of viral invasions are predictable [17], the relatively rapid breakdown of spatial synchrony at distances greater than 150 km (figure 1c) suggests that effective synchronization of interventions might be more generally operationally feasible. Future studies using GPS tracking and/or population genetics may improve understanding of how additional factors, such as roost behaviour (e.g. blood shared meals), bat culling, livestock density and deforestation influence bat dispersal and consequently the spatial scale of viral maintenance and disease control interventions [4,10].

RVNAs in apparently healthy bats are generally thought to arise from 'abortive' infections which do not lead to transmission or death [27]; however, the long-term fate of these individuals in the wild and their epidemiological significance for rabies transmission at the population level was unknown. Our results show extended recapture intervals for formerly seropositive bats and shorter recapture intervals for bats that remained seropositive suggesting enhanced survival of bats after rabies exposures and relatively rapid waning immunity, respectively (figure 3; electronic supplementary material, figure S8). One limitation of our analysis was that some seroconversions may have been missed, particularly for bats with long recapture intervals. However, the main conclusions from these results are unlikely to be sensitive to these gaps. First, positive–negative–positive histories could explain the less frequently observed long positive-to-positive recapture intervals. Individuals in these long intervals were evidently not common enough to change the observed distribution in the positive-to-positive class and this was still the shortest seroprevalence class (0.63 years). Second, in principle negative–positive–negative seroconversions could be hidden in negative-to-negative class distribution, however, if entire cycles of negative–positive–negative commonly happened, we would expect the negative-to-negative distribution to be extended, which was not the case. Indeed, the duration of positive-to-negative seroconversions were on average longer than any other category (figure 3a). Additional exposures while seropositive (e.g. immune-boosting) could also extend the positive-to-negative distribution, which would further support our conclusion on protective immunity. Together, the potentially protective effect of abortive infections and our direct observations of antibody waning confirm the prediction of earlier mathematical models that waning protective immunity contributes to the viability of spatial maintenance despite low viral prevalence [10]. Future quantitative exercises that incorporate uncertainty from missed events (e.g. hidden Markov models) would help to estimate true rates of antibody waning and seroconversion.

To our knowledge, our results are a rare example where dynamics of infection in a wild bat reservoir can be linked to observed rates of spillover to non-bat species. Here, the ability to derive these relationships arose from the unusually high frequency of bat-to-livestock transmission. Although we found consistent positive relationships between seroprevalence in bat populations and the intensity of rabies spillover to livestock, relationships were weak. Weak associations are partly explained by the difficulty of inferring periods of viral shedding from serological data in wildlife. However, relationships did not strengthen even at larger spatio-temporal scales where viral shedding and seroprevalence might be expected to be more consistent (electronic supplementary material, figure S10). One caveat of studying serology is that does not record active infection, it records past exposures that did not lead to active infection (in the case of rabies). In theory, direct data of viral circulation (e.g. virus detection in saliva in vampire bats via RT-PCR) could more accurately describe risk but such data are precluded by the low prevalence of active rabies infection in wild bats [26]. Further, nearly all variance explained was attributable to random effects, which suggests there are some missing important predictors. This exercise exemplifies the modern perception of

spillover as a complex ecological process, where several conditions need to align and part of these occur at the level of spillover hosts [1]. Data on exposure rates and livestock immunity might further help predict VBRV spillovers to livestock.

In summary, our results provide new lines of evidence for spatial-mediated viral maintenance, weak density dependence and protective waning immunity in vampire bat rabies. Improved understanding of how the configuration of bat colonies and bat dispersal interact to shape rabies incidence and spatial spread in bats is, therefore, a vital next step to improve rabies management in Latin America. Our study also showed the challenge of linking accessible, longitudinal measures of viral circulation in the reservoir to spillovers; suggesting that more holistic approaches that include complementary data from bat reservoirs and variation arising within spillover hosts will be necessary to predict pathogen emergence.

Ethics. Capture and sampling methods were approved by the Research Ethics Committee of the University of Glasgow School of Medical, Veterinary and Life Sciences (Ref081/15) and by the University of Georgia Animal Care and Use Committee (A2014 04-016-Y3-A5). Collection and exportation permits were approved by the Peruvian government (428-2008-INRENA-IFFS-DCB, RD-0299-2010-AG-DGFFS-DGEFFS, RD-273-2012-AG-DGFFS-DGEFFS, 003851-AG-DGFFS, 004692-AG-DGFFS, 005216-AG-DGFFS, 008977-AG-DGFFS, 011989-AG-DGFFS, RD-009-2015-SERFOR-DGGSPFFS, RD-264-2015-SERFOR-DGGSPFFS, RD-142-2015-SERFOR-DGGSPFFS, RD-054-2016-SERFOR- DGGSPFFS).

Data accessibility. Model tables and code are available online in the Zenodo repository (https://doi.org/10.5281/zenodo.5013655) [45]. This information is open access.

Electronic supplementary material is available online [46].

Authors' contributions. D.K.M.: data curation, formal analysis, investigation, methodology, software, visualization, writing—original draft, writing—review and editing; N.M.: formal analysis, methodology, software, writing—review and editing; A.B.: data curation, methodology, writing—review and editing; C.T.: methodology, writing—review and editing; W.V.: methodology, project administration, writing—review and editing; S.R.: methodology, project administration, writing—review and editing; J.E.C.: methodology, writing—review and editing; C.S.: project administration, writing—review and editing; N.F.: project administration, writing—review and editing; M.V.: formal analysis, investigation, methodology, supervision, validation, writing—original draft, writing—review and editing; D.G.S.: conceptualization, data curation, formal analysis, funding acquisition, investigation, methodology, project administration, supervision, validation, writing—original draft, writing—review and editing.

All authors gave final approval for publication and agreed to be held accountable for the work performed therein.

Conflict of interest declaration. We declare we have no competing interests.

Funding. This work was funded by the Wellcome Trust (Sir Henry Dale Fellowship: 102507/Z/13/A; Senior Research Fellowship: 217221/Z/19/Z) and the US National Science Foundation (DEB-1020966). D.K.M. was funded by the Human Frontier Science Program (RGP0013/2018) and the Mexican National Council for Science and Technology (CONACYT: 334795/472296).

Acknowledgements. We thank SERFOR for permissions for fieldwork. We thank John Claxton for assistance with sample storage and permits. We thank Megan Griffiths and Laura Bergner for helpful feedback on initial versions of the manuscript.

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
