## [Peer Review File · Proceedings of the Royal Society B: Biological Sciences]

Review History

RSPB-2021-1676.R0 (Original submission)

Review form: Reviewer 1 (John Drake)

Recommendation

Major revision is needed (please make suggestions in comments)

Do you have any concerns about statistical analyses in this paper? If so, please specify them explicitly in your report.

Yes

It is a condition of publication that authors make their supporting data, code and materials available - either as supplementary material or hosted in an external repository. Please rate, if applicable, the supporting data on the following criteria.

Is it accessible?

Yes

Is it clear?

Yes

Is it adequate?

Yes

Comments to the Author

Submitted to PRSB

The submitted work reports on a study seeking to link rabies virus transmission in vampire bats to spillover risk in humans and livestock. The system is a fascinating ecological situation and the authors are commended for the strength of their long-term study. It is nonetheless difficult to discern whether the scientific objectives of the work are achieved due to two weaknesses with the current manuscript. The first could be readily addressed in a revision. The second concerns some of the statistical analysis and it is unclear whether this can be addressed or not. I describe my concerns about these two aspects of the manuscript and conclude with some minor comments for the authors to consider.

The first weakness is largely stylistic. As written, the manuscript lacks a set of clearly articulated aims, hypotheses, or questions. This makes it difficult to assess whether or not the goals of the paper are in fact achieved (indeed, it is unclear to me whether they are achievable with the amount of data that is available). Besides this, there are a large number of issues with the writing that interfere with communication. I think these can be attended to by a close and critical reading of the manuscript. The majority of these concern word choice, noun-verb agreement, and phrasing. I have not documented all of the concerns I identified, but for illustration will list those that are found in the abstract. Similar issues are found throughout the manuscript

- line 30: this is an odd use of the verb "predicate"; I suggest rephrasing
- line 34: perhaps you mean "estimated" rather than "approximated"; although often used interchangeable in ordinary discourse, these words have specific meanings in technical writing that (I think) are not consistent with the authors' intent
- line 35: you write that the "dynamics" are correlated, but I think that what you mean is that some observable (e.g. incidence) is correlated among sites
- line 37: missing a comma
- line 40: environmental effects can't be "supported" but perhaps these observations "provide evidence of environmental effects" or "supported the hypothesis that..."
- line 42: "limited successes of bat culling" is awkward phrasing, I suggest something with "the limited success that has been achieved by culling..."

The second issue concerns the gamm model which is the core analysis reported here. In brief, I am concerned that the model is too flexible and therefore unreliable. For instance, a delta AIC of two is pretty small for such high-dimensional models. But you have 24 such models -- which are virtually all equivalent! I worry that having 24 models all clustered together is actually resulting from the fact that the statistical models is essentially unidentifiable. Of course, this is not surprising given the use of a semiparametric model and random effects. How many parameters does the model have (including hyper-parameters of the basis functions, random effects, etc.) compared with available degrees of freedom? You attribute the large confidence intervals to a small sample size, but I wonder if it has more to do with model specification? This would also explain the poor predictive performance. Perhaps you could start with some simpler models and work up to something that is more complicated. Related, predictive performance for spillover is quantified using marginal and conditional pseudo- R^2 , but it is not clear what would count as a "good" prediction. I don't think these numbers can possibly get close to one for all practical purposes. I don't have any intuition for what the reported values (0.06 and 0.48) mean in practice. Related, if the point is to understand the predictive performance of the model, it is important to validate the model with data withheld from the model fitting process.

Minor issues

- line 140: It is unclear where the evidence for 93.7% similarity comes from. If this is a known statistic, please provide a reference. Otherwise, documenting this should be part of the methods and results. (i.e. what samples were doubly tested to arrive at this statistic?)

- lines 178-194: I think I understand what the authors are attempting to do here, but it does have the flavor of "calculating statistics on statistics". I'm not sure that I understand how these statistical tests will behave under these circumstances. Can the authors provide some intuition? For instance, when is the mantel test expected to be significant? Is it possible that the authors could perform simulations to show that these statistical approaches would indeed detect the hypothesized pattern when in fact present (and not vice versa)?

- lines 197-222: I think information is missing from this description of methods. Which covariates were modeled nonparametrically? What kind of basis function? How were interactions handled? It's not clear to me whether the model described in the first paragraph was actually fit or if that paragraph is a generic explanation and the second paragraph actually describes the methodology. How does the exploratory analysis avoid overfitting? I'm not questioning that the analysis was done correctly (it's clearly a sophisticated use of gams), but I don't follow all the logic and modeling details.

- line 250: please provide citation for post hoc Dunn test as this is not familiar

- I don't see how the seroconversion study accounts for a case that converted and then reconverted (which would then look like no change of status). Wouldn't you need something like a hidden Markov model to model this properly? Maybe the authors have some time scale argument to suggest these "double conversions" (either pos-neg-pos or neg-pos-neg) would be so rare as to be negligible. But, if this is the case then I missed it. Please explain.

- Within-cluster spatial synchrony: are these correlation coefficients reported anywhere, perhaps as heat maps? If yes, I missed this.

- Discussion: I don't really see how this study provides evidence one way or another to address the effectiveness of culling as an intervention. This should either be made more clear or removed from the discussion.

In closing, although I know that I have been critical, I commend the authors for what has clearly been a very labor-intensive and challenging study.

Sincerely,
John Drake

Review form: Reviewer 2

Recommendation

Major revision is needed (please make suggestions in comments)

Scientific importance: Is the manuscript an original and important contribution to its field?

Acceptable

General interest: Is the paper of sufficient general interest?

Acceptable

Quality of the paper: Is the overall quality of the paper suitable?

Acceptable

Is the length of the paper justified?

Yes

Should the paper be seen by a specialist statistical reviewer?

No

Do you have any concerns about statistical analyses in this paper? If so, please specify them explicitly in your report.

No

It is a condition of publication that authors make their supporting data, code and materials available - either as supplementary material or hosted in an external repository. Please rate, if applicable, the supporting data on the following criteria.

Is it accessible?

Yes

Is it clear?

Yes

Is it adequate?

Yes

Do you have any ethical concerns with this paper?

No

Comments to the Author

The authors present the results of a valuable long-term study on rabies in vampire bats focused on: 1) drivers of seroprevalence dynamics, 2) seroconversion and waning antibodies, 3) relationships between bat seroprevalence and livestock outbreaks. The monitoring of 39 colonies over 11 years with 427 recaptured individuals makes this a valuable contribution to the literature. I thought the authors did a good job of presenting the data and analyses. However, at the moment, the paper does not have strong conclusions that are well-supported by the data. This may be rectified with some additional context or restructuring of the manuscript, but in its present form it seems more appropriate to a specialist journal.

Main points:

Drivers of bat rabies seroprevalence: The authors search over a large landscape of covariates and models, despite this there are no strong predictors except spatial clustering and random effects that do not lend themselves to much biological insight (Fig. 2). The discussion of these results (lns 364 - 388) stretches to provide some biological conclusions. The main result of decaying synchrony with distance, and what this implies for persistent circulation, could use some additional support and/or discussion. Fig. 1c and 2a suggest some local clustering, but the degree of asynchrony at larger spatial scales is less clear. As a counter-point, Fig. 1b suggests widespread outbreaks in 2011 and 2013. As a result, should the interpretation of Fig 1c instead be that there is a lack of power to detect any correlation at larger scales (grey n.s. effects) rather than actual asynchrony. Correlations at local scales would seem to be a given for any infectious disease. Given the sampling shown in Fig 1b, I think Fig. 2b should be changed to be more transparent about what is extrapolated, perhaps by using solid lines for times when sampling was occurring and maybe grey and/or dotted lines for extrapolations.

Seroconversions and antibody decay: As a wildlife disease expert, but not a bat or virus expert I felt like this component was a potential strong suit of the paper for which more could be done. I'm guessing that these sorts of longitudinal data are rare or even unique to this project, but as a non bat or rabies expert I need some additional context to motivate the importance of this work. For the titer loss analyses, it appears that the analyses were based on a subset of paired tests for which the titers declined. I wonder if the data should be further thinned to include only those that have high titers to start with (presumed positive exposure). This would exclude initial titers that are already low, but decreased even further as this may just be random testing variation on repeatedly unexposed individuals. Some additional text is needed on the titer cut-off used to qualify as positive since the data do not seem very bi-modal in Fig. 3b.

Bat seroprevalence and livestock outbreaks. The relationship in Fig. 4b seems weak even though it is statistically significant. The authors note this in the discussion Ln 430, but then I'm not sure what the take home conclusion is other than disease spillover is complicated.

I don't feel strongly about this, but the current paper has 3 parts that don't build on one another very much. They could be split apart and further developed. For example, the authors use the asynchrony results in #1 to conjecture about the persistence of VBRV, but the connection is weak. This conclusion could be made stronger by connecting this with a modeling exercise that demonstrates persistence at some spatial scale and a correlogram similar to the one observed with that sampling effort. Perhaps this model could also include the titer loss information and thus tie together #1 and #2 a little better.

Minor points:

Ln 463. "Prevailing paradigms": As a disease ecologist, but not rabies bat expert, I still need these explained to me.

Could/should the Mantel tests be run on the least-cost distance rather than euclidean distance?

Line 172-175. Can you describe your reasoning behind using 10 and 20km cutoffs?

Line 401: What established effects? Livestock and bat populations positively correlated?

Ln 160. Livestock density was determined by Gridded Livestock of the World. Please provide some context here on why this is a good estimate and single year 2010 is useful across the 11yr study. I'm guessing the livestock are the dominant large mammal and resource for vampire bats in this region, but I'm not sure that is explicitly stated in the manuscript.

The figures are very good and show the data well.

Decision letter (RSPB-2021-1676.R0)

05-Oct-2021

Dear Miss Meza:

I am writing to inform you that your manuscript RSPB-2021-1676 entitled "Ecological determinants of rabies virus dynamics in vampire bats and spillover to livestock" has, in its current form, been rejected for publication in Proceedings B.

This action has been taken on the advice of referees, who have recommended that substantial revisions are necessary. With this in mind we would be happy to consider a resubmission, provided the comments of the referees are fully addressed. However please note that this is not a provisional acceptance.

4) Data - please see our policies on data sharing to ensure that you are complying (<https://royalsociety.org/journals/authors/author-guidelines/#data>).

Sincerely,
Professor Hans Heesterbeek
mailto: proceedingsb@royalsociety.org

Associate Editor
Board Member: 1

Comments to Author:

Both reviewers agreed that this long-term analysis is important to understand reservoir dynamics and spillover events. However, they also pointed out significant problems arising from uneven data sampling over a long period of time (11 years). Both reviewers found that the manuscript needs a better framework including clear aims and hypotheses. Reviewer #1 highlighted concerns related to the statistical models that should be addressed in future versions. Reviewer #2 indicated that certain parts of the manuscript required additional context, especially for non-experts in bat rabies.

Reviewer(s)' Comments to Author:

Referee: 1

Comments to the Author(s)

Submitted to PRSB

The submitted work reports on a study seeking to link rabies virus transmission in vampire bats to spillover risk in humans and livestock. The system is a fascinating ecological situation and the authors are commended for the strength of their long-term study. It is nonetheless difficult to discern whether the scientific objectives of the work are achieved due to two weaknesses with the current manuscript. The first could be readily addressed in a revision. The second concerns some of the statistical analysis and it is unclear whether this can be addressed or not. I describe my concerns about these two aspects of the manuscript and conclude with some minor comments for the authors to consider.

The first weakness is largely stylistic. As written, the manuscript lacks a set of clearly articulated aims, hypotheses, or questions. This makes it difficult to assess whether or not the goals of the paper are in fact achieved (indeed, it is unclear to me whether they are achievable with the amount of data that is available). Besides this, there are a large number of issues with the writing that interfere with communication. I think these can be attended to by a close and critical reading of the manuscript. The majority of these concern word choice, noun-verb agreement, and phrasing. I have not documented all of the concerns I identified, but for illustration will list those that are found in the abstract. Similar issues are found throughout the manuscript

- line 30: this is an odd use of the verb "predicate"; I suggest rephrasing

- line 34: perhaps you mean "estimated" rather than "approximated"; although often used interchangeable in ordinary discourse, these words have specific meanings in technical writing that (I think) are not consistent with the authors' intent

- line 35: you write that the "dynamics" are correlated, but I think that what you mean is that some observable (e.g. incidence) is correlated among sites

- line 37: missing a comma

- line 40: environmental effects can't be "supported" but perhaps these observations "provide evidence of environmental effects" or "supported the hypothesis that..."

- line 42: "limited successes of bat culling" is awkward phrasing, I suggest something with "the limited success that has been achieved by culling..."

The second issue concerns the gamm model which is the core analysis reported here. In brief, I am concerned that the model is too flexible and therefore unreliable. For instance, a delta AIC of two is pretty small for such high-dimensional models. But you have 24 such models -- which are virtually all equivalent! I worry that having 24 models all clustered together is actually resulting from the fact that the statistical models is essentially unidentifiable. Of course, this is not surprising given the use of a semiparametric model and random effects. How many parameters does the model have (including hyper-parameters of the basis functions, random effects, etc.) compared with available degrees of freedom? You attribute the large confidence intervals to a small sample size, but I wonder if it has more to do with model specification? This would also explain the poor predictive performance. Perhaps you could start with some simpler models and work up to something that is more complicated. Related, predictive performance for spillover is quantified using marginal and conditional pseudo- R^2 , but it is not clear what would count as a "good" prediction. I don't think these numbers can possibly get close to one for all practical purposes. I don't have any intuition for what the reported values (0.06 and 0.48) mean in practice. Related, if the point is to understand the predictive performance of the model, it is important to validate the model with data withheld from the model fitting process.

Minor issues

- line 140: It is unclear where the evidence for 93.7% similarity comes from. If this is a known statistic, please provide a reference. Otherwise, documenting this should be part of the methods and results. (i.e. what samples were doubly tested to arrive at this statistic?)
- lines 178-194: I think I understand what the authors are attempting to do here, but it does have the flavor of "calculating statistics on statistics". I'm not sure that I understand how these statistical tests will behave under these circumstances. Can the authors provide some intuition? For instance, when is the mantel test expected to be significant? Is it possible that the authors could perform simulations to show that these statistical approaches would indeed detect the hypothesized pattern when in fact present (and not vice versa)?
- lines 197-222: I think information is missing from this description of methods. Which covariates were modeled nonparametrically? What kind of basis function? How were interactions handled? It's not clear to me whether the model described in the first paragraph was actually fit or if that paragraph is a generic explanation and the second paragraph actually describes the methodology. How does the exploratory analysis avoid overfitting? I'm not questioning that the analysis was done correctly (it's clearly a sophisticated use of gams), but I don't follow all the logic and modeling details.
- line 250: please provide citation for post hoc Dunn test as this is not familiar
- I don't see how the seroconversion study accounts for a case that converted and then reconverted (which would then look like no change of status). Wouldn't you need something like a hidden Markov model to model this properly? Maybe the authors have some time scale argument to suggest these "double conversions" (either pos-neg-pos or neg-pos-neg) would be so rare as to be negligible. But, if this is the case then I missed it. Please explain.
- Within-cluster spatial synchrony: are these correlation coefficients reported anywhere, perhaps as heat maps? If yes, I missed this.
- Discussion: I don't really see how this study provides evidence one way or another to address the effectiveness of culling as an intervention. This should either be made more clear or removed from the discussion.

In closing, although I know that I have been critical, I commend the authors for what has clearly been a very labor-intensive and challenging study.

Sincerely,
John Drake

Referee: 2

Comments to the Author(s)

The authors present the results of a valuable long-term study on rabies in vampire bats focused on: 1) drivers of seroprevalence dynamics, 2) seroconversion and waning antibodies, 3) relationships between bat seroprevalence and livestock outbreaks. The monitoring of 39 colonies over 11 years with 427 recaptured individuals makes this a valuable contribution to the literature. I thought the authors did a good job of presenting the data and analyses. However, at the moment, the paper does not have strong conclusions that are well-supported by the data. This may be rectified with some additional context or restructuring of the manuscript, but in its present form it seems more appropriate to a specialist journal.

Main points:

Drivers of bat rabies seroprevalence: The authors search over a large landscape of covariates and models, despite this there are no strong predictors except spatial clustering and random effects that do not lend themselves to much biological insight (Fig. 2). The discussion of these results (lns 364 - 388) stretches to provide some biological conclusions. The main result of decaying synchrony with distance, and what this implies for persistent circulation, could use some additional support and/or discussion. Fig. 1c and 2a suggest some local clustering, but the degree of asynchrony at larger spatial scales is less clear. As a counter-point, Fig. 1b suggests widespread outbreaks in 2011 and 2013. As a result, should the interpretation of Fig 1c instead be that there is a lack of power to detect any correlation at larger scales (grey n.s. effects) rather than actual asynchrony. Correlations at local scales would seem to be a given for any infectious disease. Given the sampling shown in Fig 1b, I think Fig. 2b should be changed to be more transparent about what is extrapolated, perhaps by using solid lines for times when sampling was occurring and maybe grey and/or dotted lines for extrapolations.

Seroconversions and antibody decay: As a wildlife disease expert, but not a bat or virus expert I felt like this component was a potential strong suit of the paper for which more could be done. I'm guessing that these sorts of longitudinal data are rare or even unique to this project, but as a non bat or rabies expert I need some additional context to motivate the importance of this work. For the titer loss analyses, it appears that the analyses were based on a subset of paired tests for which the titers declined. I wonder if the data should be further thinned to include only those that have high titers to start with (presumed positive exposure). This would exclude initial titers that are already low, but decreased even further as this may just be random testing variation on repeatedly unexposed individuals. Some additional text is needed on the titer cut-off used to qualify as positive since the data do not seem very bi-modal in Fig. 3b.

Bat seroprevalence and livestock outbreaks. The relationship in Fig. 4b seems weak even though it is statistically significant. The authors note this in the discussion Ln 430, but then I'm not sure what the take home conclusion is other than disease spillover is complicated.

I don't feel strongly about this, but the current paper has 3 parts that don't build on one another very much. They could be split apart and further developed. For example, the authors use the asynchrony results in #1 to conjecture about the persistence of VBRV, but the connection is weak. This conclusion could be made stronger by connecting this with a modeling exercise that demonstrates persistence at some spatial scale and a correlogram similar to the one observed with that sampling effort. Perhaps this model could also include the titer loss information and thus tie together #1 and #2 a little better.

Minor points:

Ln 463. "Prevailing paradigms": As a disease ecologist, but not rabies bat expert, I still need these explained to me.

Could/should the Mantel tests be run on the least-cost distance rather than euclidean distance?

Line 172-175. Can you describe your reasoning behind using 10 and 20km cutoffs?

Line 401: What established effects? Livestock and bat populations positively correlated?
Ln 160. Livestock density was determined by Gridded Livestock of the World. Please provide some context here on why this is a good estimate and single year 2010 is useful across the 11yr study. I'm guessing the livestock are the dominant large mammal and resource for vampire bats in this region, but I'm not sure that is explicitly stated in the manuscript.

The figures are very good and show the data well.

Author's Response to Decision Letter for (RSPB-2021-1676.R0)

See Appendix A.

RSPB-2022-0860.R0

Review form: Reviewer 3

Recommendation

Major revision is needed (please make suggestions in comments)

Scientific importance: Is the manuscript an original and important contribution to its field?

Excellent

General interest: Is the paper of sufficient general interest?

Good

Quality of the paper: Is the overall quality of the paper suitable?

Acceptable

Is the length of the paper justified?

Yes

Should the paper be seen by a specialist statistical reviewer?

No

Do you have any concerns about statistical analyses in this paper? If so, please specify them explicitly in your report.

Yes

It is a condition of publication that authors make their supporting data, code and materials available - either as supplementary material or hosted in an external repository. Please rate, if applicable, the supporting data on the following criteria.

Is it accessible?

N/A

Is it clear?

N/A

Is it adequate?

N/A

Do you have any ethical concerns with this paper?

No

Comments to the Author

I'd like to congratulate the authors for working and presenting results from such an extensive dataset. This manuscript is indeed a significant contribution to the knowledge of the spillover of RABV from bats to livestock.

However, some issues may be addressed prior to the acceptance of the manuscript.

The main issues are the lack of characterization (sex, age, etc) of the individuals found in each roost prior to the joining of smaller colonies and the least-cost distance analysis, possibly leading to biased results.

Lines 121-3: Joining smaller colonies into a cluster without prior characterization of the individuals might not be appropriate, since the risk of infection may be different between males and females, young and adults, etc.

Lines 123-4: The sentence "As nearby sites would be expected to have more similar observations of rabies exposure..." is presented as an assumption, but lacks evidence. Please elucidate or reformulate.

Lines 125: Although least-cost distances is an interesting technique to group sites based on the Euclidean distances, two problems emerge: (1) Euclidean distances ignore possible geographical barriers between them (as aforementioned in the manuscript). In a country with a rugged terrain such as Peru, this is certainly something that should be addressed. Please refer to DOI: 10.1093/jmammal/gyz177 and (2) this relationship between different roosts may depend on the sex and age of the individuals. For example, individuals from bachelor roosts may not meet with bachelors from other roosts, but in the other hand may frequently meet with females from harems. Similarly, females may not meet females from other harem.

Lines 138-53: the average values for a 10 km radius (~8 sq km) may not express the geographic suitability of the roost itself, specially for elevation and terrain ruggedness. Why not use the elevation of the site itself? As for the ruggedness, why not calculate the mean slope of the terrain around the site. The TRI as presented is confusing. As for the LCD, please refer to the comments above.

Line 154: Please provide the reference for the VBRV spread velocity in Peru.

Lines 191-5: Please refer to all previous comments and verify if unexpected results from GAMM may not be caused by biased parametrization of your variables. I really think this last effort to improve the parametrization may be beneficial to your manuscript.

Results were not evaluated since these issues may be addressed first.

Review form: Reviewer 4**Recommendation**

Accept with minor revision (please list in comments)

Scientific importance: Is the manuscript an original and important contribution to its field?

Excellent

General interest: Is the paper of sufficient general interest?

Good

Quality of the paper: Is the overall quality of the paper suitable?

Good

Is the length of the paper justified?

Yes

Should the paper be seen by a specialist statistical reviewer?

Yes

Do you have any concerns about statistical analyses in this paper? If so, please specify them explicitly in your report.

No

It is a condition of publication that authors make their supporting data, code and materials available - either as supplementary material or hosted in an external repository. Please rate, if applicable, the supporting data on the following criteria.

Is it accessible?

N/A

Is it clear?

N/A

Is it adequate?

N/A

Do you have any ethical concerns with this paper?

No

Comments to the Author

Meza et al describe a longitudinal analysis of bats and Rabies in Peru. They did an impressive work in time span, time-consuming field/cave sampling and in the total amount of bats sampled. The transmission dynamics of viruses in wild life are always challenging to explain. They aimed to resolve some difficult questions: do environmental factors modulate baseline risk for VBRV transmission, how within-host processes affect population-level dynamics and if seroprevalence correlates with epidemiological cycles and most importantly, with spillover events. They studied the patterns of virus transmission using an 11-year, spatially-replicated sero-survey of 3,709 Peruvian vampire bats and co-occurring outbreaks in livestock. They used a generalized additive mixed model of seroprevalence that showed no influence of demographic or environmental factors. In addition, they showed long-term survival following rabies exposure and antibody waning, supporting hypotheses that immunological mechanisms influence viral maintenance in bat populations. Surprisingly, they found that seroprevalence in bats was only weakly correlated with outbreaks in livestock and once again, this reinforces the challenge of spillover prediction. Their manuscript is well written and after the past revision, it was importantly improved. Some minor points need to be revised but publication is recommended.

Abstract:

I do recommend to change the last sentence "Successful management of vampire bat rabies requires improved understanding of viral transmission within networks of bat colonies." Many times it was discussed that this kind of research performed by them was needed to understand virus transmission, this great effort leads to a some other catchy last phrase. What do you suggest for successful management? A more down to earth last phrase is better.

General

They sample size and year span is impressive.

Spatial correlation of outbreaks showing through bat serology how outbreaks are clustered in small distances – something that was shown from livestock data, but not from bat data.

The opportunity to actually proof that rabies immunity wanes in wild vampire bats, and they could survive infections, really cool!

Seroprevalence did not predict rabies outbreaks in bovines, which it is not a big surprise.

Introduction

L54-64: In general, I didn't like the tone of the first paragraph, where they are seeing bats as culprits of pathogen emergence. It gives the impression that they are to blame for zoonotic events to happen, which is a dangerous message to give for bat conservation.

L111: please elaborate how antibody presence will serve as a "proxy". Whether here or somewhere else in the manuscript.

Methods

L117: how abundant or frequent was to find "clean" *Desmodus* colonies or were they mixed? Could this have influenced the sampling and results?

L121-123: So, if you had more than 5 individuals, and the colonies were closer than 10km, they were considered independent? What was the minimal distance between sites? And the minimal distance between the 11 clusters? What was the rationale behind considering this value of 10km as a deciding factor?

Viral detection and phylogenetics in bats and livestock might have shown some light in the assumption of correlation of outbreaks. Why did you do not aimed to test this? Please explain.

L130-133: how are both methods comparable? Do cutoffs were the same or similar?

L134: explain how was the cutoff defined

L135-136: why is seropositivity a reflection of exposure ≤ 6 months? Please elaborate.

L140-141: the hypotheses should be explained briefly in the main text and not as supplementary

L152-153: I don't understand what does "then are effectively infinite" refers to.

For the spatial synchrony, shouldn't you consider the clusters (N=11) instead? Or this this how the authors are identifying the clusters? It is not clear to me is these colonies are considered independent points, because the authors haven't specified the distance between them. Bats could be moving between colonies frequently (especially males).

L162: SENASA (acronym in Spanish) should be written down for non-Spanish-speaking readers.

L164-203: I am a wild life infectious diseases expert and not so much into statistical analyses and mathematical models; therefore, I was not able to revise this section in depth. If the models were the correct ones for measuring this or that for example.

L197-201: From the previous paragraph (L193), it looks like site was the only random effect evaluated, but in here you are saying that you tested for more random effects.

L213-217: these 2 sentences are confusing. Please consider to re write them more clearly.

L229-231: citation needed

Results

L267: explain "deviance" briefly and in this context

L291-296: Their results are only interpretable in terms of antibodies. What are the effects of amplification of the virus by bat density and age structure, but these bats are dying from infection, so they are biased on sampling serology on the bats that survived the event. This is an alternative explanation that the one presented in their discussion (L357-360).

L310: "bats that never seroconverted" seems confusing. Please re-phrase it.

Discussion

L379-382. Is this true in your system as well, are the colonies closer to livestock rearing bigger? I suspect that culling campaigns (and other unofficial practices of population control) are biased to livestock rearing areas, so in many cases you might actually have smaller colony sizes. So, I am not sure if cattle density is a good proxy of bat population size.

L380-386: is livestock vaccinated in this area?

L387-388: suggest why they decline.

L396: if prolonged antibody responses, but then they waned...related to food abundance? Seems a bit confusing, please change it.

L441-442: any in vivo studies of virus shedding? Or longitudinal research in bats colonies regarding this? If this would have been detected, would you have expected different results?

L445-446: do you mean virus shedding or virus detection in saliva? Do shedding means transmission? Maybe write as "virus detection in saliva"

L463-464: "build and improve disease management": identifying but -understanding- these patterns might do this. Please re-phrase it.

Figures:

Figure 3 b is too busy and gray lines are almost impossible to discern. Consider depicting it differently (maybe as S7?)

Supplemental material:

There are a couple of typos

I recommend including table S1 as a main text table. Table 1. Valuable information is written down here that will make it easier for the reader for understanding.

Decision letter (RSPB-2022-0860.R0)

08-Jul-2022

Dear Miss Meza:

Your manuscript has now been peer reviewed and the reviews have been assessed by an Associate Editor. The reviewers' comments (not including confidential comments to the Editor) and the comments from the Associate Editor are included at the end of this email for your reference. As you will see, the reviewers and the Associate Editor have raised some concerns with your manuscript and we would like to invite you to revise your manuscript to address them.

When submitting your revision please upload a file under "Response to Referees" - in the "File Upload" section. This should document, point by point, how you have responded to the reviewers' and Editors' comments, and the adjustments you have made to the manuscript. We also require a copy of the revised manuscript showing track changes to be uploaded.

Research ethics:

Use of animals and field studies:

It is a condition of publication that data supporting your paper are made available either in the electronic supplementary material. Authors must complete the 'data accessibility' section in the submission system. This should list the database and accession number for all data from the article that has been made publicly available, for instance:

NB. From April 1 2013, peer reviewed articles based on research funded wholly or partly by RCUK must include, if applicable, a statement on how the underlying research materials – such as data, samples or models – can be accessed.

[http://datadryad.org/submit?journalID=RSPB&manu=\(Document not available\)](http://datadryad.org/submit?journalID=RSPB&manu=(Document not available)) which will take you to your unique entry in the Dryad repository. If you have already submitted your data to dryad you can make any necessary revisions to your dataset by following the above link.

Please include the Dryad DOI in the Data Accessibility section and reference in the paper's bibliography.

Please see our Data Sharing Policies (<https://royalsociety.org/journals/authors/author-guidelines/>).

Please submit a copy of your revised paper within three weeks. If we do not hear from you within this time your manuscript will be rejected. If you are unable to meet this deadline please let us know as soon as possible, as we may be able to grant a short extension.

Thank you for submitting your manuscript to *Proceedings B*; we look forward to receiving your revision. If you have any questions at all, please do not hesitate to get in touch.

Best wishes,
Professor Hans Heesterbeek
mailto:proceedingsb@royalsociety.org

Associate Editor

Comments to Author:

Both reviewers positively evaluated the manuscript and highlighted that the dataset and findings will greatly expand our knowledge about spillover and viral dynamics. However, both reviewers also pointed out areas that can be further improved. Please revise the manuscript and address each of their points. One of the reviewers indicated some concerns about the GAMM analyses and parametrization. The second reviewer had more general comments and suggested clarifications.

Reviewer(s)' Comments to Author:

Referee: 3

Comments to the Author(s).

I'd like to congratulate the authors for working and presenting results from such an extensive dataset. This manuscript is indeed a significant contribution to the knowledge of the spillover of RABV from bats to livestock.

However, some issues may be addressed prior to the acceptance of the manuscript.

The main issues are the lack of characterization (sex, age, etc) of the individuals found in each roost prior to the joining of smaller colonies and the least-cost distance analysis, possibly leading to biased results.

Lines 121-3: Joining smaller colonies into a cluster without prior characterization of the individuals might not be appropriate, since the risk of infection may be different between males and females, young and adults, etc.

Lines 123-4: The sentence "As nearby sites would be expected to have more similar observations of rabies exposure..." is presented as an assumption, but lacks evidence. Please elucidate or reformulate.

Lines 125: Although least-cost distances is an interesting technique to group sites based on the Euclidean distances, two problems emerge: (1) Euclidean distances ignore possible geographical barriers between them (as aforementioned in the manuscript). In a country with a rugged terrain such as Peru, this is certainly something that should be addressed. Please refer to DOI: 10.1093/jmammal/gyz177 and (2) this relationship between different roosts may depend on the sex and age of the individuals. For example, individuals from bachelor roosts may not meet with bachelors from other roosts, but in the other hand may frequently meet with females from harems. Similarly, females may not meet females from other harem.

Lines 138-53: the average values for a 10 km radius (~8 sq km) may not express the geographic suitability of the roost itself, specially for elevation and terrain ruggedness. Why not use the elevation of the site itself? As for the ruggedness, why not calculate the mean slope of the terrain around the site. The TRI as presented is confusing. As for the LCD, please refer to the comments above.

Line 154: Please provide the reference for the VBRV spread velocity in Peru.

Lines 191-5: Please refer to all previous comments and verify if unexpected results from GAMM may not be caused by biased parametrization of your variables. I really think this last effort to improve the parametrization may be beneficial to your manuscript.

Results were not evaluated since these issues may be addressed first.

Referee: 4

Comments to the Author(s).

Meza et al describe a longitudinal analysis of bats and Rabies in Peru. They did an impressive work in time span, time-consuming field/cave sampling and in the total amount of bats sampled. The transmission dynamics of viruses in wild life are always challenging to explain. They aimed to resolve some difficult questions: do environmental factors modulate baseline risk for VBRV transmission, how within-host processes affect population-level dynamics and if seroprevalence correlates with epidemiological cycles and most importantly, with spillover events. They studied the patterns of virus transmission using an 11-year, spatially-replicated sero-survey of 3,709 Peruvian vampire bats and co-occurring outbreaks in livestock. They used a generalized additive mixed model of seroprevalence that showed no influence of demographic or environmental factors. In addition, they showed long-term survival following rabies exposure and antibody waning, supporting hypotheses that immunological mechanisms influence viral maintenance in bat populations. Surprisingly, they found that seroprevalence in bats was only weakly correlated with outbreaks in livestock and once again, this reinforces the challenge of spillover prediction. Their manuscript is well written and after the past revision, it was importantly improved. Some minor points need to be revised but publication is recommended.

Abstract:

I do recommend to change the last sentence "Successful management of vampire bat rabies requires improved understanding of viral transmission within networks of bat colonies." Many times it was discussed that this kind of research performed by them was needed to understand virus transmission, this great effort leads to a some other catchy last phrase. What do you suggest for successful management? A more down to earth last phrase is better.

General

They sample size and year span is impressive.

Spatial correlation of outbreaks showing through bat serology how outbreaks are clustered in small distances - something that was shown from livestock data, but not from bat data.

The opportunity to actually proof that rabies immunity wanes in wild vampire bats, and they could survive infections, really cool!

Seroprevalence did not predict rabies outbreaks in bovines, which it is not a big surprise.

Introduction

L54-64: In general, I didn't like the tone of the first paragraph, where they are seeing bats as culprits of pathogen emergence. It gives the impression that they are to blame for zoonotic events to happen, which is a dangerous message to give for bat conservation.

L111: please elaborate how antibody presence will serve as a "proxy". Whether here or somewhere else in the manuscript.

Methods

L117: how abundant or frequent was to find "clean" *Desmodus* colonies or were they mixed? Could this have influenced the sampling and results?

L121-123: So, if you had more than 5 individuals, and the colonies were closer than 10km, they were considered independent? What was the minimal distance between sites? And the minimal distance between the 11 clusters? What was the rationale behind considering this value of 10km as a deciding factor?

Viral detection and phylogenetics in bats and livestock might have shown some light in the assumption of correlation of outbreaks. Why did you do not aimed to test this? Please explain.

L130-133: how are both methods comparable? Do cutoffs were the same or similar?

L134: explain how was the cutoff defined

L135-136: why is seropositivity a reflection of exposure <6 months? Please elaborate.

L140-141: the hypotheses should be explained briefly in the main text and not as supplementary

L152-153: I don't understand what does "then are effectively infinite" refers to.

For the spatial synchrony, shouldn't you consider the clusters (N=11) instead? Or this this how the authors are identifying the clusters? It is not clear to me is these colonies are considered independent points, because the authors haven't specified the distance between them. Bats could be moving between colonies frequently (especially males).

L162: SENASA (acronym in Spanish) should be written down for non-Spanish-speaking readers.

L164-203: I am a wild life infectious diseases expert and not so much into statistical analyses and mathematical models; therefore, I was not able to revise this section in depth. If the models were the correct ones for measuring this or that for example.

L197-201: From the previous paragraph (L193), it looks like site was the only random effect evaluated, but in here you are saying that you tested for more random effects.

L213-217: these 2 sentences are confusing. Please consider to re write them more clearly.

L229-231: citation needed

Results

L267: explain "deviance" briefly and in this context

L291-296: Their results are only interpretable in terms of antibodies. What is there are effects of amplification of the virus by bat density and age structure, but these bats are dying from infection, so they are biased on sampling serology on the bats that survived the event. This is an alternative explanation that the one presented in their discussion (L357-360).

L310: "bats that never seroconverted" seems confusing. Please re-phrase it.

Discussion

L379-382. Is this true in your system as well, are the colonies closer to livestock rearing bigger? I suspect that culling campaigns (and other unofficial practices of population control) are biased to livestock rearing areas, so in many cases you might actually have smaller colony sizes. So, I am not sure if cattle density is a good proxy of bat population size.

L380-386: is livestock vaccinated in this area?

L387-388: suggest why they decline.

L396: if prolonged antibody responses, but then they waned...related to food abundance? Seems a bit confusing, please change it.

L441-442: any in vivo studies of virus shedding? Or longitudinal research in bats colonies regarding this? If this would have been detected, would you have expected different results?

L445-446: do you mean virus shedding or virus detection in saliva? Do shedding means transmission? Maybe write as "virus detection in saliva"

L463-464: "build and improve disease management": identifying but -understanding- these patterns might do this. Please re-phrase it.

Figures:

Figure 3 b is too busy and gray lines are almost impossible to discern. Consider depicting it differently (maybe as S7?)

Supplemental material:

There are a couple of typos

I recommend including table S1 as a main text table. Table 1. Valuable information is written down here that will make it easier for the reader for understanding.

Author's Response to Decision Letter for (RSPB-2022-0860.R0)

See Appendix B.

Decision letter (RSPB-2022-0860.R1)

16-Aug-2022

Dear Dr Meza

I am pleased to inform you that your manuscript entitled "Ecological determinants of rabies virus dynamics in vampire bats and spillover to livestock" has been accepted for publication in Proceedings B.

You can expect to receive a proof of your article from our Production office in due course, please check your spam filter if you do not receive it. PLEASE NOTE: you will be given the exact page

length of your paper which may be different from the estimation from Editorial and you may be asked to reduce your paper if it goes over the 10 page limit.

Data Accessibility section

Open Access

Paper charges

Sincerely,

Professor Hans Heesterbeek

Appendix A

University of Glasgow | Institute of Biodiversity,
Animal Health & Comparative Medicine

Diana Meza

Institute of Biodiversity, Animal Health and Comparative Medicine
Graham Kerr Building, room 306
College of Medicine, Veterinary & Life Sciences
University of Glasgow, Glasgow, G12 8QQ
Tel: 00 44 (0)141 330 6626

Email: d.villa-meza.1@research.gla.ac.uk, mezadk@gmail.com

Re: [RSPB-2021-1676] entitled “Ecological determinants of rabies virus dynamics in vampire bats and spillover to livestock”.

Dear Editors of *Proceedings of the Royal Society B*,

We are pleased to submit a revised version of our manuscript for consideration to publish in *Proceedings B*. We are grateful to the editor and the referees for recognizing the importance and challenges of our long-term analysis to understand the dynamics of zoonotic viruses in bat reservoirs and the determinants of spillover events to other species. We largely agreed with the suggestions made and have thoroughly revised the manuscript. We believe these changes have strengthened the manuscript and have clarified the significance of its conclusions for managing rabies in vampire bats and more broadly for understanding the dynamics of bat viruses in their reservoirs.

Following the recommendations of the reviewers and editor, we made significant changes throughout our manuscript. In particular, we have re-written several sections to provide important context, including clear statements describing our objectives and hypotheses. We also carried out additional statistical analyses and streamlined other analyses to converge around a common narrative related to the ecological phenomena underpinning the long-term maintenance of vampire bat rabies. As a result, new figures have been added and some older figures are now presented in the supplement. In rare cases where we disagreed with reviewers’ suggestions, our reasoning for this disagreement is clearly explained.

The letter below contains a point-by-point response to each set of suggestions from the reviewers. Reviewers’ comments are provided in full and our responses are signalled in **blue** text.

We appreciate your continued consideration of this manuscript. We look forward to your response.

Kind regards,

Diana Meza

Point-by-point response to reviewer's comments

Referee 1 (John Drake)

The submitted work reports on a study seeking to link rabies virus transmission in vampire bats to spillover risk in humans and livestock. The system is a fascinating ecological situation and the authors are commended for the strength of their long-term study. It is nonetheless difficult to discern whether the scientific objectives of the work are achieved due to two weaknesses with the current manuscript. The first could be readily addressed in a revision. The second concerns some of the statistical analysis and it is unclear whether this can be addressed or not. I describe my concerns about these two aspects of the manuscript and conclude with some minor comments for the authors to consider.

The first weakness is largely stylistic. As written, the manuscript lacks a set of clearly articulated aims, hypotheses, or questions. This makes it difficult to assess whether or not the goals of the paper are in fact achieved (indeed, it is unclear to me whether they are achievable with the amount of data that is available). Besides this, there are a large number of issues with the writing that interfere with communication. I think these can be attended to by a close and critical reading of the manuscript. The majority of these concern word choice, noun-verb agreement, and phrasing. I have not documented all of the concerns I identified, but for illustration will list those that are found in the abstract. Similar issues are found throughout the manuscript

We thank the reviewer for the helpful suggestions on the manuscript and for appreciating the challenges but also the potential of long-term studies for clarifying the ecology of host-virus relationships. We apologize for the lack of clarity in our writing, particularly around the objectives of our work. We have revised the manuscript carefully and have improved the clarity, grammar, and word choice throughout. In the revised manuscript, we have largely re-written the introduction to clearly identify the unknowns surrounding the epidemiology of rabies which we sought to resolve (e.g., Lines:74-80 and 89-95), the specific aims that these knowledge gaps lead to (Lines:80-86,87-89), and how addressing these aims can improve management (e.g., Lines:96-104). To fulfil observations about the framework and composition of the manuscript we have convened to remove some analyses (e.g., within cluster synchrony and determinants of antibody waning) that no longer fit the scope of the paper expecting the streamline of our narrative has been improved.

Broadly these objectives target deeper understanding of the ecological mechanisms that enable long term maintenance of rabies. The specific objectives of the work are:

- (1) To identify the nature and geographical scale of viral maintenance, which will help to develop informed coordinated control measures.
- (2) To evaluate whether individual, population, landscape or spatiotemporal variables explain patterns of rabies exposure in bats, which we suggest could help to resolve ongoing controversy surrounding whether rabies is maintained through spatially-dynamic or local (i.e. population level) processes.
- (3) To empirically test whether viral exposures provide transient protective immunity, a previously hypothesized (but untested) mechanism for spatial rabies maintenance in vampire bats
- (4) To determine whether antibody prevalence in bats is a useful proxy for spillover risk to non-bat species.

- line 30: this is an odd use of the verb "predicate"; I suggest rephrasing.
We have used the word "underpin" instead.

- line 34: perhaps you mean "estimated" rather than "approximated"; although often used

interchangeable in ordinary discourse, these words have specific meanings in technical writing that (I think) are not consistent with the authors' intent.

We apologize for the confusion. Our original wording referred to the fact that serological data are a proxy for active infection rates (which are unobserved with antibody data alone), not to uncertainty around the seroprevalence estimates. We appreciate this meaning was not clear without context and have removed the confusing term and have used "seroprevalence" or "antibody dynamics" instead.

- line 35: you write that the "dynamics" are correlated, but I think that what you mean is that some observable (e.g. incidence) is correlated among sites.

We used the term "dynamics" to indicate changes in seroprevalence are correlated through time, but we agree it was unclear in the text. We have instead used "fluctuations of seroprevalence" or "patterns of observed seroprevalence" throughout the text when referring to this analysis.

- line 37: missing a comma

- line 40: environmental effects can't be "supported" but perhaps these observations "provide evidence of environmental effects" or "supported the hypothesis that..."

- line 42: "limited successes of bat culling" is awkward phrasing, I suggest something with "the limited success that has been achieved by culling..."

We are sorry for the grammar mistakes, word choices and phrasing structure. They have been checked and revised throughout the new version.

The second issue concerns the gamm model which is the core analysis reported here. In brief, I am concerned that the model is too flexible and therefore unreliable. For instance, a delta AIC of two is pretty small for such high-dimensional models. But you have 24 such models -- which are virtually all equivalent! I worry that having 24 models all clustered together is actually resulting from the fact that the statistical models is essentially unidentifiable. Of course, this is not surprising given the use of a semiparametric model and random effects. How many parameters does the model have (including hyper-parameters of the basis functions, random effects, etc.) compared with available degrees of freedom? You attribute the large confidence intervals to a small sample size, but I wonder if it has more to do with model specification? This would also explain the poor predictive performance. Perhaps you could start with some simpler models and work up to something that is more complicated.

We appreciate the input on the GAMM modelling. After careful consideration of your comments and those of Reviewer 2, we have retained the GAMMs in our revised analysis as we believe it remains the most appropriate approach for our data and questions. However, we have carried out comprehensive additional checks and added explanations for our decisions. We made the following modifications and clarifications:

- (1) Changes to our site clustering rules: Given the importance of the year by cluster interaction for our hypothesis around viral maintenance and the significance of this variable in terms of deviance explained, we reconsidered the calculation of the least cost distance which determined how bat sites were clustered in the GAMM. Our new least cost distance model was based on a recent analysis of barriers to host/virus gene flow in vampire bats (Griffiths et al., 2022) which suggested that movement costs increase linearly (rather than exponentially as we previously assumed) with elevation until 3600m, after that point, movement is impossible (explained in methods Lines: 152-153,). With this new measure, we identified eleven clusters as opposed to thirteen in the previous version. The rationale of the variable cluster is explained in the main text (methods Lines: 123-126) and the methods to obtain the variable are in supplementary materials (Supplementary Text 1b, Figures S3-S5). Results using the new cluster definition remain broadly similar to the previous version of the

manuscript, with the year by cluster interaction consistently being the most informative variable.

- (2) Model flexibility/smoothing terms. We appreciate the concern associated with the flexibility of the model arising from smoothed effects in GAMMs; however, we were cautious to ensure that smoothed effects were included only where justified. We followed the procedure recommended by Simon Wood and Pedersen et al. (Pedersen et al., 2019; Wood, 2017). Briefly, smoothers were selected using a model containing all variables of interest with all continuous variables smoothed. Variables with effective degrees of freedom > 1 were retained as smoothers; the remaining were modelled as linear effects. We have included an explanation of this procedure in the methods (Lines: 180-187). We have clarified in the main text which variables were fit as linear, smoothers or random effects (Methods Lines: 186-190).
- (3) Model specification/simpler models. Our model selection procedure explored all combinations of fixed effects (i.e. including very simple models with single fixed effects). We identified the most parsimonious set of fixed effects among 10,240 nested models using the MuMin function *dredge* (Lines: 198-201). Crucially, simpler models with fewer fixed effects performed worse than our final selected model. We have made this clearer also by showing some of the bottom models in the model selection figure (now Figure S6). We also considered alternative random effects structures, but these also performed more poorly in terms of AIC and deviance explained (Methods Lines: 199-200, Supplementary Materials Table S3). Finally, to determine whether the smoothed non-linear effects were needed, we tested simpler Generalized Linear Mixed Models, however, these models presented singularities and failed to converge. In contrast, the GAMM was able to be fit and captured non-linear fluctuations of the seroprevalence among nearby bat colonies. Altogether, the diminished or intractable performance of simpler models suggests that our use of a complex model with smoothed non-linear effects is justified.
- (4) The reviewer is correct that our top models (now 13) are effectively statistically equivalent; indeed, this was the reason we mentioned all models within 2 AIC values. However, crucially the top models are also effectively biologically equivalent. Nearly all explanatory power in competitive models arises from the same 3 variables: the interaction between year and spatial cluster, month, the number of nearby rabies outbreaks in livestock, and the random effect of site. The remaining differences among competitive models arise from substitutable variables which consistently explained negligible deviance. Thus, we interpret this outcome as a confirmatory result which shows the robustness of the variables selected by our model selection approach. We clarify our interpretation of the equivalence of the top models in the results (Lines: 374-376).
- (5) Deviance explained. We agree that 28% of deviance explained may not seem much at first. However, given the ecological complexity of the system and the fact that three variables alone explained most of the deviance, we believe that the results are robust and important.
- (6) Sample size vs degrees of freedom. The sample size was of 3975 and the degrees of freedom (including hyper-parameters of the basis functions) of the full initial GAMM were 87.3. The degrees of freedom of the top model were 75.35, with residual degrees of freedom = 3899.65. We considered there should not be an evident constraint to fit the model given the large sample size versus the effective degrees of freedom. The information of the degrees of freedom is now included in the supplementary materials for the full initial GAMM (Table S3) and in the main text for the top model (Line 276).

Related, predictive performance for spillover is quantified using marginal and conditional pseudo- R^2 , but it is not clear what would count as a "good" prediction. I don't think these numbers can possibly get close to one for all practical purposes. I don't have any intuition for what the reported values (0.06 and 0.48) mean in practice. Related, if the point is to understand the predictive

performance of the model, it is important to validate the model with data withheld from the model fitting process.

We apologize for the confusion here, which we suspect arises from our insufficient explanation of the underlying rationale for this analysis in the earlier version of the manuscript. We agree that a truly predictive model (i.e., to be used for forecasting) would require more than bat seroprevalence and livestock density (e.g., measures of livestock vaccination and nearby rabies incidence) and would require hold out data for validation. Here, we merely sought to determine if there was any relationship between the two to understand the value of seroprevalence data for understanding rabies spillover. We believe this is a valid question given the immunological significance of serological data in this system. It is also an important question, given that serological data are typically more accessible than data on acute viral infections (e.g., PCR or virus isolation). However, we appreciate that our wording around 'prediction' did not convey that intention and have adjusted our wording throughout the manuscript to focus on associations and correlations.

We have clarified in the methods that we use pseudo- R^2 as a measure of goodness of fit rather to assess a good estimation for prediction. We also added a reference on the use of the conditional R^2 and marginal R^2 and for use of the delta method, which can calculate the goodness of fit for Poisson distributed generalized linear mixed models (Lines: 239-240). Given the conditional R^2 represents the goodness of fit of the full model (including random effects site and individual identifier per seroprevalence observed) and the marginal R^2 represents the goodness of fit of only fixed effects (seroprevalence in bats and livestock density), we thought it would be informative to show both (Results Lines: 315-317). The conclusion of this analysis is that serological data do capture some signal of spillover as would be expected for a proxy of recent viral circulation but fall far short of what would be required to explain or predict outbreaks. We suggest this arises from the complex nature of rabies virus maintenance in vampire bats. Specifically, the apparently spatially-mediated maintenance and waning protective immunity from non-lethal exposures that we show in other parts of the manuscript means that antibodies are a reasonable proxy for viral exposure in bats, but a relatively weak metric of spillover risk. Given that we were not aiming for a forecasting model, we opted against validating performance with a test set. A better explanation of our rationale for these results is now provided in the discussion (Lines:437-447).

Minor issues

- line 140: It is unclear where the evidence for 93.7% similarity comes from. If this is a known statistic, please provide a reference. Otherwise, documenting this should be part of the methods and results. (i.e. what samples were doubly tested to arrive at this statistic?).

This similarity was derived from a validation exercise in a previous methods paper. The reference has been added at the end of the sentence (line 150).

- Lines:178-194: I think I understand what the authors are attempting to do here, but it does have the flavor of "calculating statistics on statistics". I'm not sure that I understand how these statistical tests will behave under these circumstances. Can the authors provide some intuition? For instance, when is the mantel test expected to be significant? Is it possible that the authors could perform simulations to show that these statistical approaches would indeed detect the hypothesized pattern when in fact present (and not vice versa)?

To address the reviewer's concern, we created a null expectation for patterns of spatial synchrony that might be observed in our data. Specifically, we iteratively randomized the observed seroprevalence values by site and year, calculated synchrony and repeated the correlogram. We have amended the methods (Lines:172-174) and adapted the correlogram figure (Figure 1c). This showed that the observed levels of synchrony would not be expected by chance.

- Lines:197-222: I think information is missing from this description of methods. Which covariates

were modeled nonparametrically? What kind of basis function? How were interactions handled? It's not clear to me whether the model described in the first paragraph was actually fit or if that paragraph is a generic explanation and the second paragraph actually describes the methodology. How does the exploratory analysis avoid overfitting? I'm not questioning that the analysis was done correctly (it's clearly a sophisticated use of gams), but I don't follow all the logic and modeling details.

As described above, we have revised the methods in the manuscript to clarify our choices and approaches to model structure, model selection and implementation of non-linear and linear variables. Briefly, we tried to avoid overfitting by exploring simpler models (e.g., GLMM) and different structures of the initial model (e.g., single variable or no random effects) as explained above. A table and a figure with some of the different model structures we tested is now shown in the supplementary materials (Table S3, figure S6). Moreover, before performing model selection, we performed a thorough selection of which variables to include as smoothers (see reply first comment Table S2). Among the variables included as smoothers, we did not specify the number of knots since 'mgcv' uses the minimum number of knots required to fit the data to prevent overfitting. Unfortunately, it was not possible to completely guarantee that our model is not overfit, however, we used a robust pipeline to avoid it and observed consistent results across different model structures which provides confidence in our conclusions.

- line 250: please provide citation for post hoc Dunn test as this is not familiar.
We have added the citation, now line 246.

- I don't see how the seroconversion study accounts for a case that converted and then reconverted (which would then look like no change of status). Wouldn't you need something like a hidden Markov model to model this properly? Maybe the authors have some time scale argument to suggest these "double conversions" (either pos-neg-pos or neg-pos-neg) would be so rare as to be negligible. But, if this is the case then I missed it. Please explain.

We agree that missing seroconversions are possible between recaptures, however, we believe the most interesting conclusions of our results are unlikely to be sensitive to these gaps. First, the recaptures that remained seropositive are important indicators of the potential duration of immunity. The median time between successive seropositive captures was 0.63 years, which was the shortest of any of the possible seroprevalence states. Positive-negative-positive histories could explain the less frequently observed long positive to positive recapture intervals. However, these longer (and less trustworthy) intervals were evidently not common enough to keep the observed distribution of recapture intervals from being significantly shorter than for any other possible seroprevalence event. Second, the duration of positive to negative seroconversions is interesting because, extended intervals relative to bats that were presumably never exposed to rabies (negative to negative) would suggest enhanced survival, arising from protection from future rabies infections. Indeed, we observed that positive-negative seroconversions happened in longer times (median = 1.33 years) than any of the other categories. While in principle true histories of negative-positive-negative could also extend this distribution, if entire cycles of positive to negative were common, this would also be expected to extend the observed negative to negative distribution, which was not the case (Figure 3). Additional exposures while seropositive (e.g., boosting) could also extend the positive to negative distribution, but this would further support our conclusion on protective immunity. To clarify this analysis, we have changed the figure to violin plots with the distribution points to facilitate the visualization of the different densities (Figure 3). We also added text to the discussion to clarify our interpretation of these results (Lines: 419-433).

We appreciate the suggestion to consider a Hidden Markov Model. We agree this would be a valuable quantitative exercise to estimate true rates of antibody waning and seroconversion that incorporate uncertainty from missed events. However, here we are concerned with the qualitative

aims of demonstrating that antibody waning occurs and that the low levels of antibodies routinely observed for lyssaviruses in bats are likely to protect against future viral exposures. For this reason, we opted against the HMM but have noted that this could be a useful approach for future studies in the discussion (433-435).

- Within-cluster spatial synchrony: are these correlation coefficients reported anywhere, perhaps as heat maps? If yes, I missed this.

We have removed the within-cluster spatial synchrony analysis since was only tangentially linked to the manuscript after our revisions to the introduction. We also think it is unlikely sufficiently powered to find a real pattern.

- Discussion: I don't really see how this study provides evidence one way or another to address the effectiveness of culling as an intervention. This should either be made more clear or removed from the discussion.

We agree that the links to culling were insufficiently explained in the previous version. In the current introduction we have explained that under the hypothesis of within-colony viral maintenance via density dependent transmission (which remains pervasive among some stakeholders in Latin America, particularly those charged with management), culling might be expected to be an effective management tool; however, under the alternative hypothesis of spatially-mediated maintenance, culling would be expected to be less effective (Lines:98-104). In supporting the latter explanation for patterns of viral circulation within bats, our study raises doubts on the mechanistic basis for culling bats as a rabies management strategy. Of course, we are aware the effectiveness of culling is not directly addressed in our work, so we have been cautious in our discussion of this topic.

In closing, although I know that I have been critical, I commend the authors for what has clearly been a very labor-intensive and challenging study.

We appreciate the detailed and constructive nature of the suggestions. We hope the changes made are sufficient to address your concerns.

Referee: 2

The authors present the results of a valuable long-term study on rabies in vampire bats focused on: 1) drivers of seroprevalence dynamics, 2) seroconversion and waning antibodies, 3) relationships between bat seroprevalence and livestock outbreaks. The monitoring of 39 colonies over 11 years with 427 recaptured individuals makes this a valuable contribution to the literature. I thought the authors did a good job of presenting the data and analyses. However, at the moment, the paper does not have strong conclusions that are well-supported by the data. This may be rectified with some additional context or restructuring of the manuscript, but in its present form it seems more appropriate to a specialist journal.

Main points:

Drivers of bat rabies seroprevalence: The authors search over a large landscape of covariates and models, despite this there are no strong predictors except spatial clustering and random effects that do not lend themselves to much biological insight (Fig. 2). The discussion of these results (Ins 364 - 388) stretches to provide some biological conclusions.

We thank the reviewer for pointing out that the take home message of our manuscript was not resonating clearly. Given that reviewer 1 had a similar issue, we have largely re-written the introduction and discussion of the manuscript to clarify our objectives and conclusions. We have also removed some analyses (e.g., within cluster synchrony and determinants of antibody waning) that no longer fit the scope of the paper with a hope to streamline our narrative. The current manuscript

revolves around evaluating lines of evidence underlying different possible explanations for the long-term maintenance of rabies in bats, namely single population maintenance governed by density dependence versus spatially-mediated maintenance involving multiple populations. We have made greater efforts throughout to explain how our results relate to that broader goal. For example, the fact that spatiotemporal variables (e.g., the year by cluster interaction) are the best predictors of viral exposure, rather than proxies for bat density (e.g., livestock densities), local landscape (e.g., elevation) or bat demography (e.g., sex) is more consistent with spatial maintenance. Similarly, the dynamics of antibody loss which we observed in recaptured bats verifies previously untested prediction of a mathematical model which argued for spatial viral maintenance in this system (Blackwood et al., 2013). Our aims have been clarified at the end of the introduction (Lines: 106-112).

The main result of decaying synchrony with distance, and what this implies for persistent circulation, could use some additional support and/or discussion. Fig. 1c and 2a suggest some local clustering, but the degree of asynchrony at larger spatial scales is less clear. As a counter-point, Fig. 1b suggests widespread outbreaks in 2011 and 2013. As a result, should the interpretation of Fig 1c instead be that there is a lack of power to detect any correlation at larger scales (grey n.s. effects) rather than actual asynchrony. Correlations at local scales would seem to be a given for any infectious disease. The lack of support for asynchrony at larger distances was shared by the first reviewer. To strength our results, we created a null expectation using a randomization from the data set (see response to Reviewer's 1 9th comment above). We agree that correlation at local scales would seem to be a given, however the spatial scale of that synchrony will vary among different diseases. We were especially interested in investigating the spatial scale of rabies maintenance, since rabies is thought to be spatial maintained across different regional cycles. We understand we could be missing some synchrony patterns due to the lack of some data in some years and sites; however, we think this would be unlikely given the biology of the vampire bats (home ranges don't tend to be larger than 40 km) and the peaking of seroprevalence at different times between different clusters shown in the updated Figure 2a. We hope the reframing and explanation of rabies ecology throughout the introduction makes the motivation of the analysis clearer. We also expanded the discussion on this result (Lines:338-346,400-407).

Given the sampling shown in Fig 1b, I think Fig. 2b should be changed to be more transparent about what is extrapolated, perhaps by using solid Lines:for times when sampling was occurring and maybe grey and/or dotted Lines:for extrapolations.

We have followed the recommendation and plot the effect of the interaction year-cluster as different panel per cluster with the corresponding partial residuals. See new Figure 2a

Seroconversions and antibody decay: As a wildlife disease expert, but not a bat or virus expert I felt like this component was a potential strong suit of the paper for which more could be done. I'm guessing that these sorts of longitudinal data are rare or even unique to this project, but as a non-bat or rabies expert I need some additional context to motivate the importance of this work. For the titer loss analyses, it appears that the analyses were based on a subset of paired tests for which the titers declined. I wonder if the data should be further thinned to include only those that have high titers to start with (presumed positive exposure). This would exclude initial titers that are already low, but decreased even further as this may just be random testing variation on repeatedly unexposed individuals. Some additional text is needed on the titer cut-off used to qualify as positive since the data do not seem very bi-modal in Fig. 3b.

We thank the reviewer for the observation on how rare such longitudinal antibody data are, in particular for wildlife. As part of the reframing of the manuscript we made the following changes to contextualize this section of the work:

- (1) Additional context. We have expanded the introduction to include background on the implications of antibody waning (Lines 83-84) that is consistent with the main aim of our research of a deeper understanding of the ecological mechanisms that enable long term maintenance of rabies. Briefly, since rabies does not seem to persist within single bat colonies (e.g., bat dispersion is necessary for long-term maintenance), loss of immunoprotection would be necessary for rabies re-invasion, yet the immunoprotection role of antibodies and their natural waning in wild bats has not been empirically proved. We then discuss further discuss the implications of our results in the Discussion (Lines: 414-435).
- (2) Titer loss analysis. We decided to drop the analysis of the effect of environmental variables on antibody waning. We agree it is an important topic and should be considered in the future; however, unfortunately it no longer fits within the more focused scope of the manuscript.
- (3) Figure 3b. We apologize for the lack of appropriate description. Defining antibody titre cut-offs is challenging for any system and even more so in those with low titre responses such in vampire bat transmitted rabies, where bats can have non-lethal exposures that help build immunoprotection (Turmelle et al., 2010). However, in a previous article, we compared our serological test to gold-standard methods and we found 0.166 IU/mL to be the value that increases sensitivity and specificity to detect seropositive individuals with low-titre values (Meza et al., 2020). We have added the value of the cut-off in the figure description and a clarification in the methods (Lines: 133-135).

Bat seroprevalence and livestock outbreaks. The relationship in Fig. 4b seems weak even though it is statistically significant. The authors note this in the discussion Ln 430, but then I'm not sure what the take home conclusion is other than disease spillover is complicated.

We understand the take-home message of this figure seems simple and overgeneralizing. However, given the immunological significance of serological data in wildlife surveillance and serological data are typically more accessible than molecular surveillance data (i.e., active infection), we believe understanding the value of seroprevalence data to monitor risk is valid. Consequently, we wanted to include this analysis in our manuscript. We show that even with a large dataset this is not quite feasible to use seroprevalence to monitor future spillovers since the correlation was not strong. Further, more information should be collected to predict outbreak risk such as livestock exposure rates. We have expanded the discussion (Lines: 441-452) and conclusion of this result (Discussion, Lines: 455-458).

I don't feel strongly about this, but the current paper has 3 parts that don't build on one another very much. They could be split apart and further developed. For example, the authors use the asynchrony results in #1 to conjecture about the persistence of VBRV, but the connection is weak. This conclusion could be made stronger by connecting this with a modeling exercise that demonstrates persistence at some spatial scale and a correlogram similar to the one observed with that sampling effort. Perhaps this model could also include the titer loss information and thus tie together #1 and #2 a little better.

We revised the hypotheses of the study and decided to consolidate them under the topic of a deeper understanding of the ecological mechanisms that enable long term maintenance of rabies. We agree that a more complex modelling exercise should eventually be done to understand the ecological determinants of rabies spatial spread, however such model would require more information of the vampire bat biology that we don't have at the moment, such as bat movement data and colony distributions. The importance of gathering these data for future studies has been highlighted as an outcome of our manuscript (Lines:356-362).

Minor points:

Ln 463. "Prevailing paradigms": As a disease ecologist, but not rabies bat expert, I still need these explained to me.

We have restructured the introduction to clarify the main hypotheses about vampire bat rabies maintenance, their importance for management (Lines: 96-104) and how they are addressed by our study with specific aims (Lines: 107-112).

Could/should the Mantel tests be run on the least-cost distance rather than euclidean distance?

The aim of our analysis was to clarify the geographical spatial scale of rabies persistence, which we thought was best understood in the biologically interpretable units of Euclidian distance. This could also be more informative for designing synchronized control measures. We clarified this in the methods (Lines:165-170) and added the implications of the knowledge of geographic scales on control measures to the Introduction (Lines: 98,104)

Line 172-175. Can you describe your reasoning behind using 10 and 20km cutoffs?

Given past evidence of rabies persistence been driven by bat dispersal, we wanted to create variables that would account for the risk of rabies invading the bat colonies and we used the confirmed livestock outbreaks as proxy of this risk (i.e., if a studied bat colony experienced recent rabies invasion, then this could lead to outbreaks in the nearby livestock). Since evidence has suggested the speed of spread of rabies is ~ 10 km/year in a wave like manner, then we would expect the outbreaks that happened in ≤ 10 km and a year previous the bat sampling would suggest recent rabies invasions in the bat colonies. In addition, following a 10km/year spread, outbreaks that happened > 10 km and ≤ 20 km would be indicative of rabies invasions in the previous 2 years from sampling. We found an effect of the 10 km outbreaks on the bat seroprevalence but not in the 20km, possibly arising from studying larger areas which would produce more noise in the data, or by the lack of established vampire bat colonies in those areas. We now explain this in methods (Lines:153-156) and added re-wrote the associated discussion (Lines: 345-350).

Line 401: What established effects? Livestock and bat populations positively correlated?

Given vampire bats prefer to feed on large animals, and their primary food source is livestock, livestock rearing areas tend to be an indicator for larger vampire bat populations (Becker et al., 2018; Bohmann et al., 2018; Delpietro and Russo, 1996; Streicker et al., 2012) Therefore, livestock densities might be hypothesized to be linked to rabies circulation. We have improved the writing of this explanation in the discussion (Lines: 378-381).

Ln 160. Livestock density was determined by Gridded Livestock of the World. Please provide some context here on why this is a good estimate and single year 2010 is useful across the 11yr study. I'm guessing the livestock are the dominant large mammal and resource for vampire bats in this region, but I'm not sure that is explicitly stated in the manuscript.

Records of livestock densities from Peru are kept at department or district level but they don't include geographical coordinates. The Gridded Livestock of the World contained the highest resolution estimate available to calculate the livestock density around 10 km of the geographic location of the bat colonies. We have clarified this in the methods (Lines 148-149). We have also addressed the explicit statement of livestock densities as the main resource for vampire bats in the previous comment.

The figures are very good and show the data well.

Thank you.

Appendix B

School of Biodiversity,
One Health &
Veterinary Medicine

Diana Meza

School of Biodiversity, One health and
Veterinary Medicine
University of Glasgow, Glasgow, G12 8QQ
Tel: 00 44 (0)141 330 6626

Email: d.villa-meza.1@research.gla.ac.uk,
mezadk@gmail.com

Re: [RSPB-2022-0860] entitled “Ecological determinants of rabies virus dynamics in vampire bats and spillover to livestock”.

Dear Editors of *Proceedings of the Royal Society B*,

We are pleased to submit a revised version of manuscript RSPB-2022-0860 for consideration for publication in *Proceedings B*. We are grateful for the editor and the referees for acknowledging the importance of our study of the dynamics of viruses in bat reservoirs and the determinants of spillover events to other species. We believe the changes made in response to the reviewers’ suggestions have clarified and improved the manuscript.

Following the recommendations of the reviewers and editor, we added text to clarify our hypotheses, methods, discussion, and supplementary information. We also carried out additional descriptive statistical analyses suggested by the reviewers and added new figures and tables to the supplementary information. In all cases, these additional analyses strengthen our earlier conclusions and analytical decisions. In rare cases where we disagreed with reviewers’ suggestions, our reasoning for this disagreement is explained.

The letter below contains a point-by-point response to each set of suggestions from the reviewers. Reviewers’ comments are provided in full and our responses are signalled in **blue** text.

We appreciate your continued consideration of this manuscript. We look forward to your response.

Kind regards,

Diana Meza

Reviewer(s)' Comments to Author:

Referee: 3

Comments to the Author(s).

I'd like to congratulate the authors for working and presenting results from such an extensive dataset. This manuscript is indeed a significant contribution to the knowledge of the spillover of RABV from bats to livestock.

However, some issues may be addressed prior to the acceptance of the manuscript.

The main issues are the lack of characterization (sex, age, etc) of the individuals found in each roost prior to the joining of smaller colonies and the least-cost distance analysis, possibly leading to biased results.

We thank the reviewer for the helpful suggestions on the manuscript. In the revised manuscript, we have clarified the reviewer's two central concerns:

- 1) Individuals characterization. We ran some tests to show that the sex and age ratio in joined sites did not differ (lines 144-145, table S1).
- 2) Definition of our least-cost distance metric. We noted that we did not use Euclidean distances to compute this variable but accept that our wording on that point may not have been clear in the earlier version of the manuscript. We rewrote the definition for this metric and now explained that we computed the least-cost distances based on a resistance surface model based on elevation, which assumes a linear increase of cost of movement with elevation until 3600m (the highest reported vampire bat colony in Peru), after that point movement is not possible (lines 172-175).

Detailed explanations of the changes made to the manuscript in relation to these two main points are described below in relation to specific comments.

Lines 121-3: Joining smaller colonies into a cluster without prior characterization of the individuals might not be appropriate, since the risk of infection may be different between males and females, young and adults, etc.

To address the reviewer's concern, we analyzed the sex and age ratios across groups of the joined sites and found that they did not differ significantly (see table below). Moreover, the 6 joined sites (poorly sampled ones, ≤ 10 bats) only have 33 out of the 4,889 captured bats. As such, the small number of individuals involved would be unlikely to influence predictors of seroprevalence detected across the much larger dataset of ungrouped bat colonies. In the revised text, we have added the number of sites that were poorly sampled, the number of bats that were grouped, and a sentence explaining that sex ratios did not differ between those sites (lines 144-145). Also, we have added a table in Supplementary materials which shows the joined sites and their sex ratios (table S1).

Variable	Joined site code	χ^2	df	p-value	Ratio			Distance (km)
Sex	AMA2-AMA3*	1.47	1	0.23	13:16	3:0		2.55
	API3-API15*	2.14	1	0.14	48:41	7:1		0.23
	AYA14-AYA13*	0.03	1	0.86	29:42	1:0		2.58
	CUS8-CUS3*	0.001	1	0.97	48:29	5:4		2.12
	MDD134- MDD135* - MDD136*	0.73	2	0.69	26:32	2:1	1:2	4.15 ⁺
Age	AMA2-AMA3*	0.12	1	0.73	21:8:0	3:0:0		2.55
	API3-API15*	4.97	2	0.08	88:1:2	7:1:0		0.23
	AYA14-AYA13*	0.001	1	1.00	62:9:0	1:0:0		2.58
	CUS8-CUS3*	0.34	2	0.84	69:6:2	8:1:0		2.12
	MDD134- MDD135* - MDD136*	2.88	4	0.58	43:4:3	3:0:0	2:1:0	4.15 ⁺

Lines 123-4: The sentence “As nearby sites would be expected to have more similar observations of rabies exposure...” is presented as an assumption, but lacks evidence. Please elucidate or reformulate.

We apologize for the lack of context in this sentence. Given rabies spread among colonies is currently believed to be required for long term viral maintenance and that this inter-colony spread is driven by bat dispersal, which predominately occurs over relatively short distances [1], we should expect patterns of viral exposure to be more similar in nearby sites. Indeed, temporal patterns of rabies exposure appeared to be correlated in vampire bat colonies that had been shown by mark-recapture studies to be connected by bat dispersal in Belize [2]. We now cite this paper and provide additional context (lines 140-142).

Lines 125: Although least-cost distances is an interesting technique to group sites based on the Euclidean distances, two problems emerge: (1) Euclidean distances ignore possible geographical barriers between them (as aforementioned in the manuscript). In a country with a rugged terrain such as Peru, this is certainly something that should be addressed. Please refer to DOI: 10.1093/jmammal/gyz177 and (2) this relationship between different roosts may depend on the sex and age of the individuals. For example, individuals from bachelor roosts may not meet with bachelors from other roosts, but in the other hand may frequently meet with females from harems. Similarly, females may not meet females from other harem.

1) We apologize for the misunderstanding. We fully agree and as explained above, we did not measure the least-cost distance using Euclidian distances, but rather based on a resistance surface model based on elevation. The exact specification of the landscape resistance model follows from a recent analysis of barriers to host/virus gene flow in vampire bats which evaluated several competing resistance surfaces [3]. Our least-cost distance captures the spatial isolation of the sampled site considering possible geographical

barriers. Our calculation of least-cost distances is now clarified in the methods section (lines 172-176).

2) The reviewer is correct that our clustering does not account for the possibility that behavioral factors linked to sex or age might contribute to variation in the connectivity of nearby roosts, which may occur in nature. However, we would still expect that sites within the same cluster to be more epidemiologically linked than far away colonies. In other words, we expect there to be more within cluster linkage of rabies spread than between cluster linkage due to spatial autocorrelation arising from limits on bat dispersal. Even if the individual bat colonies that we studied within a cluster do not interact directly, they are likely to be epidemiologically linked via other (unsampled) colonies. Further, our regression analysis does include variables at the individual bat level, such as the sex and age of the bat, which might help discern between individual versus population effects on rabies transmission. We have added an explanatory sentence on the purpose of the cluster variable (lines 140-142) and also clarified in the discussion that variation in roost connectivity may influence inter-colony spread of rabies (lines 374-380 and line 434).

Lines 138-53: the average values for a 10 km radius (~8 sq km) may not express the geographic suitability of the roost itself, specially for elevation and terrain ruggedness. Why not use the elevation of the site itself?

We appreciate the comment from the reviewer which showed that our past nomenclature was confusing. We intended Elevation to describe the proportion of habitat near each bat roost that was also potentially suitable for roost establishment. So, a higher suitability around the site would imply more potential roosts nearby and therefore a potentially higher risk of rabies introductions. We did not intend this variable to describe the suitability of the roost itself (indeed all roosts were by definition suitable since they were occupied). We have renamed the variable to 'Suitability' and explain it in line 165. Similarly for the terrain ruggedness index, we were interested in the average changes in landscape around the site; hypothesizing that more rugged areas could either offer more roost sites or restrict bat dispersal or livestock rearing. We clarified this in line 167-168 and 230-232.

As for the ruggedness, why not calculate the mean slope of the terrain around the site. The TRI as presented is confusing.

We prefer the use of TRI because it is a metric that is specifically designed as a landscape level measure of ruggedness, which has been extensively studied and has interpretable values [4–6]. Ruggedness provides a measure of topography heterogeneity. The reviewer is correct that this is conceptually very similar to averaging slope across neighboring cells; indeed, TRI, slope, and roughness were fully correlated and so substituting slope for TRI would not have effect on our conclusions in the models of seroprevalence. We have added a correlogram in the supplementary materials to show the correlation between the computed landscape variables but retain TRI as the focal variable in the text (figure S1).

As for the LCD, please refer to the comments above.

Please see our replies above where we explain that the LCD variable is based on a resistance metric, which was constructed considering the biology of the bats.

Line 154: Please provide the reference for the VBRV spread velocity in Peru.

The citation has been added (line 177)

Lines 191-5: Please refer to all previous comments and verify if unexpected results from GAMM may not be caused by biased parametrization of your variables. I really think this last effort to improve the parametrization may be beneficial to your manuscript.

Results were not evaluated since these issues may be addressed first.

We have made our best effort to address the concerns over the parameterization of our model and explain the reasoning behind the variables used in the model.

Referee: 4

Comments to the Author(s).

Meza et al describe a longitudinal analysis of bats and Rabies in Peru. They did an impressive work in time span, time-consuming field/cave sampling and in the total amount of bats sampled. The transmission dynamics of viruses in wild-life are always challenging to explain. They aimed to resolve some difficult questions: do environmental factors modulate baseline risk for VBRV transmission, how within-host processes affect population-level dynamics and if seroprevalence correlates with epidemiological cycles and most importantly, with spillover events. They studied the patterns of virus transmission using an 11-year, spatially-replicated sero-survey of 3,709 Peruvian vampire bats and co-occurring outbreaks in livestock. They used a generalized additive mixed model of seroprevalence that showed no influence of demographic or environmental factors. In addition, they showed long-term survival following rabies exposure and antibody waning, supporting hypotheses that immunological mechanisms influence viral maintenance in bat populations. Surprisingly, they found that seroprevalence in bats was only weakly correlated with outbreaks in livestock and once again, this reinforces the challenge of spillover prediction. Their manuscript is well written and after the past revision, it was importantly improved. Some minor points need to be revised but publication is recommended.

We appreciate the positive feedback. We agree with the reviewer that some points in our manuscript required further explanation and we have made a great effort throughout the manuscript to follow the reviewer recommendations, while trying to not substantially increase the length of the manuscript.

Abstract:

I do recommend to change the last sentence "Successful management of vampire bat rabies requires improved understanding of viral transmission within networks of bat colonies." Many times it was discussed that this kind of research performed by them was needed to understand virus transmission, this great effort leads to a some other catchy last phrase. What do you suggest for successful management? A more down to earth last phrase is better.

We appreciate this comment, and we agree the past sentence seemed quite abstract, we have rewritten the last part of the abstract (lines 43-47).

"Together our results suggest that rabies maintenance requires transmission among multiple, nearby bat colonies which may be facilitated by waning of protective immunity. However, the likelihood of incursions and dynamics of transmission within bat colonies appear largely independent of bat ecology. The implications of these

results for spillover anticipation and controlling transmission at the source are discussed.”

General

They sample size and year span is impressive.

Spatial correlation of outbreaks showing through bat serology how outbreaks are clustered in small distances – something that was shown from livestock data, but not from bat data.

The opportunity to actually proof that rabies immunity wanes in wild vampire bats, and they could survive infections, really cool!

Seroprevalence did not predict rabies outbreaks in bovines, which it is not a big surprise.

We are glad that reviewer found our results interesting, and we hope part of the disease ecology community, zoologist and infectious disease experts will do so too.

Introduction

L54-64: In general, I didn't like the tone of the first paragraph, where they are seeing bats as culprits of pathogen emergence. It gives the impression that they are to blame for zoonotic events to happen, which is a dangerous message to give for bat conservation.

We agree with this comment, we thought the most extreme sentences were the first ones which we have rewritten (lines 66-68).

“Understanding the determinants of pathogen transmission within bat populations is necessary to manage health risks to human and companion animals as well as threats to bat conservation.”

L111: please elaborate how antibody presence will serve as a “proxy”. Whether here or somewhere else in the manuscript.

Serological surveillance is a proxy because it does not record active infection, but it does show past viral exposure. It is likely that this past exposure, in the case of rabies, did not lead to an active infection (if so, the bat sampled bat would be dead). As explained in comments below, rabies active transmission would be very difficult to observe (as explained below <1%) and given antibodies stay longer, serology provides the opportunity to do longitudinal rabies surveillance. We thought that this antibody surveillance, even if it does not show active infection, could have potentially informed of past exposures and helped to trace outbreaks in livestock, which eventually we showed in our results, it was not a straight correlation. We have clarified this in the introduction (lines 143-152)

Methods

L117: how abundant or frequent was to find “clean” *Desmodus* colonies or were they mixed? Could this have influenced the sampling and results?

Almost half of our sites (48 %) are *Desmodus* only sites. We have added this percentage in the methods (line 166). However, this would not have influenced sampling, since *Desmodus* are always prioritized for sampling over other species and seroprevalence of these other species were excluded from our analyses here, i.e., other species are only used as a covariate in our GAMM and do not contribute to the response variable. Although cross-species rabies exposures from non-*Desmodus* bats are possible, our GAMM analysis included a binomial variable of the presence/absence of other bat species and this variable was not retained in the top model, suggesting no systematic effects of other species on

seroprevalence in *Desmodus*. In less competitive models that included the presence/absence of bat species, this variable explained <1% of the variation in seroprevalence.

L121-123: So, if you had more than 5 individuals, and the colonies were closer than 10km, then were considered independent?

We thank the reviewer for this comment, we realized that there was a typo in this explanation. We apologize for this mistake. The correct structure is that colonies closer than 5 km with less than 10 individuals were joined. This means that colonies in the opposite situation (with more than 10 bats and within 5 km) would be considered independent. We have amended the typo (lines 139-140).

What was the minimal distance between sites?

The minimal distance between bat roosts was 0.23 km. This distance occurred between two colonies that were joined as one had less than 10 individuals sampled. For independent colonies the minimal distance was 0.51 km, but only 4 pairs of roosts had a distance ≤ 5 km. We added the information of the minimum distance in line 140 and to supplementary materials (table S1).

And the minimal distance between the 11 clusters?

To provide further spatial understanding of the clusters we calculated the geographic coordinates for the cluster centroids and the Euclidean distances between them. The minimum distance between two clusters is 56.61 km (between cluster 9 and 10, lines 149-150).

What was the rationale behind considering this value of 10km as a deciding factor?

We agree the rationale to join sites within 5km (note 10km was a typo as explained above) is slightly arbitrary. The decision was mainly for statistical purposes given some roosts did not have enough sample size to perform a robust analysis. We convened this grouping epidemiologically appropriate since vampire bats often use multiple roosts within geographically nearby areas (<5km [7]) and previous studies have suggested epidemiological linkage of colonies separated by up to 8 km [8]. We have added a clarification in lines 140-143.

Viral detection and phylogenetics in bats and livestock might have shown some light in the assumption of correlation of outbreaks. Why did you do not aimed to test this? Please explain.

We agree that a phylogenetic analysis of rabies viruses in bats and nearby livestock would offer interesting insights, but unfortunately, the incidence of rabies in wild bat populations is too low to be a practically useful source of viral genetic material. For example, as explained above, our earlier survey of Peruvian vampire bats detected rabies virus in only 0.8% of saliva samples and importantly these samples are limited to only a small subset of the time series that was available in our serum collection [9]. We have previously analyzed virus sequences from livestock; however, these data do not provide direct insight into the dynamics within bat populations since they involve multiple observational filters related to reporting of outbreaks, livestock vaccination, and the possibility of livestock movement

from the locations of infection [10,11]. Some clarification on why the use of serology is needed when doing longitudinal surveillance in the introduction (lines 111-120)

L130-133: how are both methods comparable? Do cutoffs were the same or similar?

The cut off is not the same since the methodology is different (qualitative vs quantitative) but we did a correlation of the serological status between both of the methods on 2,365 serum samples and they are 93.7% similar. We have rewritten the methods for clarification (lines 159-160)

L134: explain how was the cutoff defined.

The cutoff was defined using a ROC curve, which showed that 0.166 was the cut off that maximized sensitivity and specificity. This was shown in a previously published article [12]. We added a note that explains this (lines 157-159)

L135-136: why is seropositivity a reflection of exposure <6 months? Please elaborate.

This is based on earlier experimental inoculations of bats with rabies virus, which showed that rabies virus neutralizing antibodies dropped below the detection limit by 6 months after inoculation [13,14]. We have added a moved this to the introduction and clarified the sentence (line 117-120)

L140-141: the hypotheses should be explained briefly in the main text and not as supplementary.

We have written a short paragraph that summarizes our hypothesis in lines 228-236. Unfortunately given the large number of variables considered, we do not have space to explain every possible relationship so instead refer readers to the supplementary table which contains this information. We are willing to follow the editor's discretion if they believe this (very large) table must be included in the main text.

L152-153: I don't understand what does "then are effectively infinite" refers to.

We made this in the model to simulate impossible movement beyond the biological relevant elevation (3,600 m). This means that the after the cutoff elevation (3,600m), bat movement is not allowed by our least cost distance model (i.e., these movements would have infinite costs). We have clarified the statement (line 175-76).

For the spatial synchrony, shouldn't you consider the clusters (N=11) instead? Or this this how the authors are identifying the clusters? It is not clear to me is these colonies are considered independent points, because the authors haven't specified the distance between them. Bats could be moving between colonies frequently (especially males). The aim of the synchrony analysis was to clarify the spatial scale of rabies persistence, which we thought was best understood in the biologically interpretable units of Euclidian distance. Avoiding grouping sites into clusters was essential since doing so would have eliminated all variation within clusters (i.e., at short distances), which is our main question. It is expected that bat colonies are non-independent, particularly at shorter distances, and this was what we were trying to identify with this analysis. This has been clarified in lines 191-192.

L162: SENASA (acronym in Spanish) should be written down for non-Spanish-speaking readers.

We have removed the acronym and used the written-out version of the institution to avoid confusion (line 186)

L164-203: I am a wild-life infectious diseases expert and not so much into statistical analyses and mathematical models; therefore, I was not able to revise this section in depth. If the models were the correct ones for measuring this or that for example.

We appreciate the feedback that reviewer was able to provide. Earlier reviewers gave close scrutiny to the statistical methods applied and we are confident in their suitability.

L197-201: From the previous paragraph (L193), it looks like site was the only random effect evaluated, but in here you are saying that you tested for more random effects.

Apologies for the confusion. We use a multi-step model selection approach to determine which variables should be included in our GAMM. Specifically, followed the recommendations in [15,16], where we first determine which variables should include a smoother by including all the potential non-linear variables (e.g. suitability, TRI, livestock density, etc. table S3). Then we determine the random effects structure that could improve the fit of the model. For this we tested different random effects structures (e.g., only site, site nested in cluster, etc.) but for the final model selection step where we explore all fixed effects variables, we only kept the random effects of site. We found that the model that included site alone captured most of the unexplained variance. Finally for last step, we ran a full comparison of all the potential nested models (fixed variables) where include the pre-determined random effect (i.e., site only) and smoother structures as well as all the other fixed effect variables (e.g., sex, age, etc.). We have rewritten this section for clarification (lines 222-237)

L213-217: these 2 sentences are confusing. Please consider to re write them more clearly.

We have re-written these sentences, now lines 248-249.

L229-231: citation needed

We have added the correspondent citation (line 263)

Results

L267: explain "deviance" briefly and in this context

We use the percentage of the deviance because it is the proportion of dependent variable variance accounted for by the model, which is a ratio indicating how close the fit of a model is to being perfect (difference in likelihoods of the model and the perfect model). For general purposes, we defined this in methods (line 226) that we refer to relative deviance as a measure of goodness of fit. This term might be a better understood terminology for a wider audience (epidemiologist, ecologist, etc.). We have added in the methods section

L291-296: Their results are only interpretable in terms of antibodies. What is there are effects of amplification of the virus by bat density and age structure, but these bats are dying from infection, so they are biased on sampling serology on the bats that survived the

event. This is an alternative explanation that the one presented in their discussion (L357-360).

We appreciate this observation given that our study is predominately based on bats that survived rabies exposures, and thus are a proxy for recent viral circulation in each colony. Age biases in mortality are possible, but in our dataset they were evidently not strong enough to be retained in our statistical models. For example, if juveniles were highly sensitive to rabies mortality, we might have expected lower seroprevalence in surviving juveniles compared to adults, which was not the case. Bat density while not measured here, an earlier study found no relationship between vampire bat colony size and seroprevalence. This phenomenon was explained to arise through the behavioral drivers of transmission (biting) which may be independent of bat density [1,17]. Despite these arguments, we agree with the reviewer on the limitations of serology and have acknowledged these more clearly in this section of the discussion (472-474).

L310: "bats that never seroconverted" seems confusing. Please re-phrase it.

We removed this text in brackets as we agreed it was confusing and not adding further clarification to our results (line 337).

Discussion

L379-382. Is this true in your system as well, are the colonies closer to livestock rearing bigger? I suspect that culling campaigns (and other unofficial practices of population control) are biased to livestock rearing areas, so in many cases you might actually have smaller colony sizes. So, I am not sure if cattle density is a good proxy of bat population size. Our variable inner-circle measured outbreaks around 10 km of the sites, so the livestock areas were within the foraging range of the bats. We agree that practices for population control will affect colony sizes, however colony size is not a necessary a good measure of population density as observed in previous research [1,17], areas where livestock rearing happens could comprise bat populations of multiple small colonies. We discussed cattle density as a proxy for bat density because different studies in Latin America have shown correlations between cattle and bat population numbers [18,19]. We have rewritten this paragraph in our discussion (lines 407-423)

L380-386: is livestock vaccinated in this area?

Currently this is only partially known [11]. Since vaccination coverage is not available for all the studied sites, unfortunately it could not be included. In addition, vaccination in this area is mostly a reactive measure to rabies outbreaks rather than a preventive constant measure, which would complicate the interpretation of the results.

L387-388: suggest why they decline.

See response below

L396: if prolonged antibody responses, but then they waned...related to food abundance? Seems a bit confusing, please change it.

We agree the past written structure made difficult to understand our suggested hypothesis on why low food abundance might weaken immune responses. Research in vampire bats and little red flying foxes has shown that changes in food resources alters host-pathogen

interactions [20]. Here, we suggest that the areas with higher livestock density might need less effort to forage and the physiological response could be higher investment in immune responses. We have rewritten the last section of this paragraph to clarify our hypothesis (lines 420-423).

L441-442: any in vivo studies of virus shedding? Or longitudinal research in bats colonies regarding this? If this would have been detected, would you have expected different results?

There are in-vivo shedding studies where bats shed for a few days then die and this was very dose dependent. However, the observed seroprevalence could be associated with outbreaks that occurred 6-months in the past (as suggested in the experiments made by Turmelle et.al. [13,14]). Also, the observed seroprevalence and outbreaks could be a consequence of multiple rabies introductions which limits our observation process. Given the low probability of detecting rabies in saliva, longitudinal research on viral shedding is highly challenging. We have expanded our discussion explaining the challenge of such data (lines 474-476)

L445-446: do you mean virus shedding or virus detection in saliva? Do shedding means transmission? Maybe write as “ virus detection in saliva”

We have rewritten this sentence following the reviewer’s suggestion (line 474).

L463-464: “build and improve disease management”: identifying but –understanding- these patterns might do this. Please re-phrase it.

We have rephrased the concluding paragraph of our discussion (line 482-489).

Figures:

Figure 3 b is too busy and gray lines are almost impossible to discern. Consider depicting it differently (maybe as S7?)

To make the figure clearer, we divided it into two panels: top - no change in serostatus; bottom - change in serostatus. We also have added a legend of the points and lines (figure 3b).

Supplemental material:

There are a couple of typos

We thank the reviewer for this comment, and we apologize for the typos. We have carefully checked and review the supplemental material.

I recommend including table S1 as a main text table. Table 1. Valuable information is written down here that will make it easier for the reader for understanding.

Given constraints on the length of the manuscript, we were unable to incorporate the table in the main text and instead added several sentences outlining our hypotheses. We hope these changes are sufficient to address the reviewer’s concern.

Reviews references

1. Blackwood JC, Streicker DG, Altizer S, Rohani P. 2013 Resolving the roles of immunity, pathogenesis, and immigration for rabies persistence in vampire bats. *Proc. Natl. Acad. Sci. U.S.A.* **110**, 20837–20842. (doi:10.1073/pnas.1308817110)
2. Becker DJ, Broos A, Bergner LM, Meza DK, Simmons NB, Fenton MB, Altizer S, Streicker DG. 2021 Temporal patterns of vampire bat rabies and host connectivity in Belize. *Transboundary Emerging Dis* **68**, 870–879. (doi:10.1111/tbed.13754)
3. Griffiths ME, Broos A, Bergner LM, Meza DK, Suarez NM, da Silva Filipe A, Tello C, Becker DJ, Streicker DG. 2022 Longitudinal deep sequencing informs vector selection and future deployment strategies for transmissible vaccines. *PLOS Biology* **20**, e3001580. (doi:10.1371/journal.pbio.3001580)
4. Riley S, Degloria S, Elliot SD. 1999 A terrain ruggedness index that quantifies topographic heterogeneity. *International Journal of Science* **5**, 23–27.
5. Wilson MFJ, O'Connell B, Brown C, Guinan JC, Grehan AJ. 2007 Multiscale Terrain Analysis of Multibeam Bathymetry Data for Habitat Mapping on the Continental Slope. *Marine Geodesy* **30**, 3–35. (doi:10.1080/01490410701295962)
6. Flantua SGA, O'Dea A, Onstein RE, Giraldo C, Hooghiemstra H. 2019 The flickering connectivity system of the north Andean páramos. *J Biogeogr* **46**, 1808–1825. (doi:10.1111/jbi.13607)
7. Trajano E. 1996 Movements of Cave Bats in Southeastern Brazil, with Emphasis on the Population Ecology of the Common Vampire Bat, *Desmodus rotundus* (Chiroptera). *Biotropica* **28**, 121. (doi:10.2307/2388777)
8. Becker DJ, Broos A, Bergner LM, Meza DK, Simmons NB, Fenton MB, Altizer S, Streicker DG. 2021 Temporal patterns of vampire bat rabies and host connectivity in Belize. *Transboundary and Emerging Diseases* **68**, 870–879. (doi:10.1111/tbed.13754)
9. Griffiths ME, Bergner LM, Broos A, Meza DK, Filipe A da S, Davison A, Tello C, Becker DJ, Streicker DG. 2020 Epidemiology and biology of a herpesvirus in rabies endemic vampire bat populations. *Nat Commun* **11**, 5951. (doi:10.1038/s41467-020-19832-4)
10. Benavides JA, Valderrama W, Streicker DG. 2016 Spatial expansions and travelling waves of rabies in vampire bats. *Proc. R. Soc. B.* **283**, 20160328. (doi:10.1098/rspb.2016.0328)
11. Benavides JA, Rojas Paniagua E, Hampson K, Valderrama W, Streicker DG. 2017 Quantifying the burden of vampire bat rabies in Peruvian livestock. *PLoS Negl Trop Dis* **11**, e0006105. (doi:10.1371/journal.pntd.0006105)
12. Meza DK, Broos A, Becker DJ, Behdenna A, Willett BJ, Viana M, Streicker DG. 2021 Predicting the presence and titre of rabies virus-neutralizing antibodies from low-volume serum samples in low-containment facilities. *Transbound Emerg Dis* **68**, 1564–1576. (doi:10.1111/tbed.13826)
13. Jackson FR, Turmelle AS, Farino DM, Franka R, McCracken GF, Rupprecht CE. 2008 Experimental rabies virus infection of big brown bats (*Eptesicus fuscus*). *Journal of Wildlife Diseases* **44**, 612–621. (doi:10.7589/0090-3558-44.3.612)
14. Turmelle AS, Jackson FR, Green D, McCracken GF, Rupprecht CE. 2010 Host immunity to repeated rabies virus infection in big brown bats. *Journal of General Virology* **91**, 2360–2366. (doi:10.1099/vir.0.020073-0)
15. Wood SN. 2017 *Generalized additive models: An introduction with R, second edition*. CRC Press. See <https://books.google.co.uk/books?id=HL-PDwAAQBAJ>.
16. Pedersen EJ, Miller DL, Simpson GL, Ross N. 2019 Hierarchical generalized additive models in ecology: an introduction with mgcv. *PeerJ* **7**, e6876. (doi:10.7717/peerj.6876)

17. Streicker DG *et al.* 2012 Ecological and anthropogenic drivers of rabies exposure in vampire bats: implications for transmission and control. *Proc. R. Soc. B.* **279**, 3384–3392. (doi:10.1098/rspb.2012.0538)
18. Becker DJ *et al.* 2018 Livestock abundance predicts vampire bat demography, immune profiles and bacterial infection risk. *Phil. Trans. R. Soc. B* **373**, 20170089. (doi:10.1098/rstb.2017.0089)
19. Delpietro HA, Marchevsky N, Simonetti E. 1992 Relative population densities and predation of the common vampire bat (*Desmodus rotundus*) in natural and cattle-raising areas in north-east Argentina. *Preventive Veterinary Medicine* **14**, 13–20. (doi:10.1016/0167-5877(92)90080-Y)
20. Plowright RK, Field HE, Smith C, Divljan A, Palmer C, Tabor G, Daszak P, Foley JE. 2008 Reproduction and nutritional stress are risk factors for Hendra virus infection in little red flying foxes (*Pteropus scapulatus*). *Proc. R. Soc. B.* **275**, 861–869. (doi:10.1098/rspb.2007.1260)